# Topographic controls on divide migration, stream capture, and diversification in riverine life

Nathan J. Lyons[1], Pedro Val[2], James S. Albert[3], Jane K. Willenbring[4], Nicole M. Gasparini[1]

[1]Department of Earth and Environmental Sciences, Tulane University, New Orleans, LA, USA
[2]Department of Geology, Federal University of Ouro Preto, Ouro Preto, Brazil
[3]Department of Biology, University of Louisiana at Lafayette, Lafayette, CA, USA
[4]Scripps Institution of Oceanography, University of California San Diego, La Jolla, CA, USA

*Correspondence to*: Nathan J. Lyons (nlyons@tulane.edu)

**Abstract.** Drainages reorganise in landscapes under diverse conditions and process dynamics that impact biotic distributions and evolution. We first investigated the relative control that Earth surface process parameters have on divide migration and stream capture in scenarios of base level fall and heterogeneous uplift. A model built with the Landlab toolkit was run 51,200 times in sensitivity analyses that used globally observed values. Large-scale drainage reorganisation occurred only in the model runs within a limited combination of parameters and conditions. Uplift rate, rock erodibility, and the magnitude of perturbation (base level fall or fault displacement) had the greatest influence on drainage reorganisation. The relative magnitudes of perturbation and topographic relief limited landscape susceptibility to reorganisation. Stream captures occurred more often when the channel head distance to divide was low. Stream topology set by initial conditions strongly affected capture occurrence when the imposed uplift was spatially heterogeneous.

We also integrated simulations of geomorphic and biologic processes to investigate relationships among topographic relief, drainage reorganisation, and riverine species diversification in the two scenarios described above. We used a new Landlab component called SpeciesEvolver that models species at landscape scale following macroevolutionary process rules. More frequent stream capture and less frequent stream network disappearance due to divide migration increased speciation and decreased extinction, respectively, especially in the heterogeneous uplift scenario where final species diversity was often greater than the base level fall scenario. Under both scenarios, the landscape conditions that led to drainage reorganisation also controlled diversification. Across the model trials, the climatic or tectonic perturbation was more likely in low relief landscapes to drive more extensive drainage reorganisation that in turn increased the diversity of riverine species lineages, especially for the species that evolved more rapidly. This model result supports recent research of natural systems that implicates drainage reorganisation as a mechanism of riverine species diversification in lowland basins. Future research applications of SpeciesEvolver software can incorporate complex climatic and tectonic forcings as they relate to macroevolution and surface processes, as well as region- and taxon-specific organisms based in rivers as well as those on continents at large.

# 1 Introduction

Topographic structure is primarily controlled by climate, tectonics, and lithological erodibility (Whipple, 2004; Anders et al., 2008; Han et al., 2015; Perron, 2017). Spatiotemporal variability in these controls can induce spatially variable erosion rates that can alter the planform topology of drainage networks and the longitudinal profiles of channels, in regions with only meters to thousands of meters of relief (Gilbert, 1877; Giachetta et al., 2014; Forte et al., 2016; Willett et al., 2014; Whipple et al., 2017). Drainages reorganise by divide migration that is the progressive movement of a drainage divide, and stream capture that occurs when a portion of a stream network loses connectivity to its former network as it joins an adjacent network (Fig. 1; Bishop, 1995). Climatically and tectonically induced changes to base level, water flow direction, and erosional processes can alter topographic structure and reorganise drainages in settings such as internally-draining fault-bounded basins (D'Agostino et al., 2001), precipitation gradients (Bonnet, 2009), transient passive margins (Prince et al., 2011; Moodie et al., 2018), intercontinental strike-slip faults (Guerit et al., 2016), and lateral variations in lithologic erodibility (Gallen et al., 2018; Harel et al., 2019). Yet little attention has been paid to the impact of drainage reorganisation on riverine biota despite the longstanding recognition of their interactions (Bishop, 1995; Albert et al., 2018).

Topographic relief links climatic and tectonic forcings to biological evolution (Badgley et al., 2017). High rates of uplift and erosion are found in regions with great relief and high diversity in many groups of terrestrial organisms, such as birds and mammals (Simpson, 1964; Rahbek, 1997; Grenyer et al., 2006). Relief enlarges environmental gradients and offers varied habitats, among other factors that form and maintain diversity (Badgley et al., 2017). Conversely, riverine groups, notably fish, are often most diverse in lowland basins where relief is low (Hoeinghaus et al., 2004; Muneepeerakul et al., 2008). The 'river capture hypothesis' of Albert et al. (2018) puts forth the idea that large and frequent drainage captures in lowland basins contributes to high diversity of fish in these regions. The challenges in testing this hypothesis include relating species-dense assemblages of riverine organisms to limited records of drainage reorganisation. Mechanistic models of biologic and geomorphic processes can provide information on complex process interactions, potentially guiding future empirical studies on the river capture hypothesis and other lines of inquiry on the intersection of landscapes and biodiversity.

From a macroevolutionary perspective, regional biodiversity is characterised by species richness (the number of species) in a clade (a group of organisms, e.g. a species, descending from a common ancestor) arising from the processes of speciation (species lineage splitting forming new species) and extinction (species lineage termination) (Stanley, 1979). From a biogeographic perspective, species richness in a geographically circumscribed region (e.g., island, drainage basin) is a function of speciation, extinction and dispersal (species geographic range expansion) (Hubbell, 2001), and evolutionary time (Rabosky, 2009). Species dispersal affects gene flow among populations and genetic diversity within populations, and the probability of species extinction increases when dispersal ability is limited. Long-term geographic separation of populations (i.e. allopatry) is a mechanism of speciation as populations genetically diverge due to reproductive isolation (Coyne, 1992).

Recent research implicates drainage reorganisation in the evolutionary origin and ecological maintenance of high riverine biological diversity in many regions (e.g., Waters and Wallis, 2000; Albert and Crampton, 2010; Bossu et al., 2013; Roxo et al., 2014; Craw et al., 2016; Albert et al., 2018; Gallen, 2018). In the context of drainage reorganisation, the organisms of a species can disperse across a greater area when a stream network expands by divide migration (Fig. 1; Burridge et al., 2008). Divide migration can also cause networks to shrink, which increases the likelihood of species extinction (Grant et al., 2007). Stream capture increases speciation probability and lineage diversity in riverine taxa following basin fragmentation (Burridge et al., 2006; Kozak et al. 2006; Tagliacollo et al., 2015; Waters et al., 2015; Craw et al. 2016), and lowers extinction risk following basin integration by allowing the geographic range of species to expand (Grant et al., 2007; 2010).

Computational modelling is increasingly used to investigate landscape and biological evolution, although largely separately. Landscape evolution modelling has illuminated drainage reorganization in response to tectonic strain (Castelltort et al., 2012), spatially variable bedrock erodibility (Giachetta et al., 2014), and autogenic processes (Pelletier, 2004), among other causative factors. Implementing captures in models has included probabilistic (Howard, 1971), numerical (Whipple et al., 2017) and coupled numerical-analytical (Goren et al., 2014) approaches. Models have also been used to demonstrate quantitative techniques to identify regions undergoing drainage reorganisation (Willett et al., 2014; Forte and Whipple, 2018). Meanwhile, species richness has been simulated as an output of spatially explicit ecological models that have static topography and that do not include tectonic or geomorphic processes (Gotelli et al., 2009; Rangel et al., 2018). Salles et al. (2019) used landscape evolution models to quantify the connectivity of landscape portions with implications for biodiversity. However, computational models that integrate biological evolution with numerically implemented surface processes have yet to be used in published research to our knowledge.

In this paper we first investigate the conditions and parameter space in which drainages reorganise in response to a single perturbation in modelled landscapes. We address the following questions: 'Are landscapes with low or high topographic relief more susceptible to drainage reorganisation?' and 'What process parameters influence this susceptibility for landscapes with a given relief?' These questions are explored with simulations of the surface processes most often used in a landscape evolution model (LEM), namely stream incision and hillslope diffusion. Some processes potentially important to stream capture (e.g., inter-basin groundwater flow, mass wasting) are not included in this study. We also investigate the conditions and parameter space in which the lineages of species diversify in response to topographic changes. The species represent those that live in or are closely associated with drainage networks; e.g. the organisms that are adapted to the channels, floodplains or riparian forests of streams. We integrate three macroevolutionary processes (dispersal, speciation, and extinction) into a LEM to ask: 'Do the same parameters that lead to drainage reorganisation also impact riverine species diversity within a landscape?' Investigating these three questions together allows us to associate patterns of topographic change with diversification, and

apply the new modelling tool, SpeciesEvolver. Through this investigation we provide a framework for future model-based research of the biological macroevolution that can follow the surface processes often included in LEMs.

## 2 Description of modelling tools

We built a LEM for this study using the Landlab modelling toolkit (Hobley et al., 2017; Barnhart et al., 2020). This scientific
computing software provides tools to build 2-dimensional numerical models of Earth surface dynamics. A landscape is represented by a model grid with configurable spatial dimensions that Landlab users can easily set with built-in routines. Processes are implemented as model components that control the values of fields, which are data associated with spatial elements of the grid, including a field of topographic elevation stored at grid nodes. Processes are effectively coupled when model components interact with the same fields. Landlab is open source, written in the Python programming language, and
available for download at https://landlab.github.io. Landlab version 2.0 was used in this study.

We used a new Landlab component called 'SpeciesEvolver' that enables researchers to model biological macroevolution in response to landscape change. This software evolves taxonomic objects (e.g., populations, species) at geologic, macroevolutionary, and landscape scales (Lyons et al., 2020). Each taxonomic object has at minimum a geographic range
within the model grid, macroevolutionary rules, and a lineage, all of which can be influenced by landscape properties and processes. For example, surface processes drive topographic change, which may alter habitat connectivity that in turn influences the macroevolutionary processes of the simulated taxon.

Taxa are implemented as object classes in the source code of SpeciesEvolver. The base class provides behaviour and properties
that can be expanded or overridden. Users can create classes of alternative and more complex taxa that inherit from the base class, which saves users from recoding behaviour already implemented in the software. Users may make essentially limitless modifications—some more readily implemented than others—including requiring a timeframe for an isolation period for a fragmented taxon to spawn new taxa, and probabilistic-based rule adaptions for macroevolution processes. In this study, we use the only taxon class currently distributed with SpeciesEvolver called 'ZoneTaxon'. Instances of this class are associated
with zone objects that manage the location of the taxa in the grid. The location of zones can be set using elevation ranges, landforms, or other attributes defined by the user. Our use of SpeciesEvolver for stream-based species in this study is described in Sect. 3.

## 3 Experiment design

We investigated the questions posed in Sect. 1 using a model-based experiment. Drainage reorganisation was triggered by
perturbing the simulated topography in two model scenarios: a base level fall scenario with an instantaneous drop of elevation

along one model grid boundary, and a fault throw scenario with an instantaneous block uplift of half of the model grid. We predict that major perturbation-driven topographic changes will lead to drainage reorganisation, which in turn will affect species diversification. We conducted sensitivity analyses to identify the model input variables that contributed most strongly to the variation of drainage reorganisation and species diversification as the inputs changed. The intent of these analyses is to describe key relationships among model inputs and outputs given the modelled processes and wide parameter space. Henceforth we use the term 'factor' to refer to a model input parameter that was varied in the sensitivity analyses, and the term 'response' to refer to a model output variable investigated in the analyses. Each scenario was composed of 25,600 trials, which was the number of trials necessary for the total order Sobol index (described in Sect. 3.1) to decrease below 1 % as more trials were run. The scenarios only differed by the perturbation mechanism. Each trial of a given scenario differed only by the values of the seven sensitivity analyses factors presented in Table 1 and described in Sect. 3.2.

## 3.1 Sensitivity analyses

We conducted a sensitivity analysis of each model response for both scenarios. A model response, $Y$ can be represented with the function, $f$ as

$$Y = f(X_1, \dots, X_c) \tag{1}$$

where $\{X_1, \dots, X_c\}$ is the factor set, and $c$ is the count of factors in this set. The factor sets for the experiment trials were generated using a quasi-random Sobol sequence (Sobol, 1967). This sequence distributes factor values throughout the parameter space more uniformly than a purely random sequence. The sensitivity analysis benefits from a uniformly distributed parameter space because the response is better characterised when the model is parameterised throughout the interval of all factors.

We used the variance-based Sobol (2001) sensitivity analysis method implemented in the sensitivity analysis library, SALib (Herman and Usher, 2017). Variance-based methods (1) analyse sensitivity globally throughout the parameter space rather than local methods that analyse sensitivity around a point in the parameter space, and (2) decompose the variance of a response due to variation in the model factors. The sensitivity of an output response to input factors is quantified using Sobol indices. The relative contribution of $X_i$ to the response variance is the Sobol first-order sensitivity index,

$$S_i = \frac{Var(E[Y|X_i])}{Var(Y)} \tag{2}$$

where $E[Y|X_i]$ is the conditional expectation of $Y$ given $X_i$. The first-order index does not include interaction among factors to influence the response. The second-order sensitivity index includes the interaction of $X_i$ and $X_j$ as

$$S_{ij} = \frac{Var(E[Y|X_{\setminus i}, X_{\setminus J}])}{Var(Y)} \tag{3}$$

where $X_{\setminus i}$ and $X_{\setminus j}$ are all factors excluding $X_i$ and $X_j$, respectively. The total effect of $X_i$ including interactions is the total order sensitivity index as

$$S_{Ti} = 1 - \frac{Var(E[Y|X_{\setminus i}])}{Var(Y)} \tag{4}$$

In this study we use Sobol indices to rank the relative influence that factors have on controlling model response variables under the conditions of the two scenarios. The total, first and second order Sobol indices were calculated for each response in each scenario. For a given response (e.g., topographic relief described in Sect. 3.2.1), the ranking of first order indices indicates the relative influence that each factor individually contributed to the response variance. The second order index indicates the combined influence of two factors on the response. The total order index encapsulates the total variance of the model response including first and higher order interactions. For example, a factor with a large total order index and small first order index indicates a response is influenced through higher order interaction of multiple factors. A simple way to conceptualize these indices is that they act as the percent contribution of model factors to output variance. The sum of the contributions of the total order indices can be greater than 100 % because the variances of interactions among the factors are included more than once in the summation.

## 3.2 Model trial progression

The base level fall and fault throw scenarios proceeded in the same way. Only the mechanisms that perturbed the topography differed between the two scenarios. A model grid with steady state elevation and streams seeded with species was established during the initial conditions phase. The first action in the perturb phase was either dropping the base level, or faulting the topography, depending on the scenario. The simulated landscape and lineages evolved together in this second phase until elevation returned to steady state, at which point the trial ended. In both phases of both scenarios, the time step duration was 1000 years and steady state was defined the same. Steady state was reached when changes in the mean and standard deviation of elevation over the prior 1000 time steps (or 1 million model years) was less than 1 %. A generalised trial is illustrated in Fig. 2. Trial parameters are summarised in Table 1 and described in more detail below. The factor values of each trial are provided in a data repository associated with this paper (Lyons et al., 2019).

### 3.2.1 Initial conditions phase

A Landlab raster model grid was initialised with dimensions of 10 by 20 km and a node spacing of 100 m. The left and right boundaries of the grid were closed to mass export, and the top and bottom boundaries were set to open. These boundary conditions were selected to represent a generic landscape drained by streams that dominantly flow to the north and south separated by a main divide that spanned the width of the grid. The initial topography of each trial was generated in a two-step process where first random elevation noise was generated and then topography was developed from that noise. The initial noise is necessary for streams to develop. The noise was generated using a pseudorandom number generator that set the initial elevations of grid nodes to values between 0 and 1 m. At each grid node the generator selected a number randomly by performing operations on a previously generated value. The first number generated was computed using a seed value that acted

as the initial internal state of the random number generator. The value of the seed for each trial was set by the sensitivity analysis factor, 'initial elevation seed' that varied between the arbitrary values of 1 and 20,000 among the trials.

The topography of the model grid evolved from the initial generated noise to steady state during the initial conditions phase. The grid elevation field was updated in each 1000-year time step. The land surface elevation, $z$ (m) at each node was modelled following detachment-limited fluvial incision using the stream power model (Howard et al., 1994) and linear hillslope diffusion (Culling, 1963). The downslope transport of hillslope material is proportional to the gradient of the local land surface multiplied by transport coefficient, $k_d$ (m$^2$ yr$^{-1}$). The change in elevation over time, $t$ (yr) at each node was modelled as

$$\frac{\delta z}{\delta t} = U - KA^m S^n + k_d \nabla^2 z \qquad (5)$$

where $U$ (m yr$^{-1}$) is rock uplift rate relative to base level, $A$ (m$^2$) is contributing drainage area as a surrogate for discharge where we used a uniform precipitation rate of 1 m yr$^{-1}$, $S$ (m m$^{-1}$) is local channel slope, and $m$ and $n$ were constants in this experiment. Base level in this study was the top and bottom boundaries of the model grid. The erosion coefficient, $K$ (m$^{1-2m}$ yr$^{-1}$) encapsulates surface erodibility, and in real landscapes it is commonly assumed to be influenced by rock strength, channel width and bed material, and runoff among other variables (Whipple and Tucker, 1999).

Previously published values of empirically observed uplift, stream incision, and diffusion parameters guided the selection of factor intervals that were explored in experiment trials (Table 1). Regional rock uplift was simulated at each time step by uniformly increasing the elevation of all grid nodes except the nodes along the grid boundary, which were not changed. The magnitude of uplift rate in each trial was set by the 'uplift rate' sensitivity analysis factor and varied generally from orogenic to cratonic values. The maximum value of this factor was $1 \times 10^{-3}$ m yr$^{-1}$, which is slightly lower than the rapid uplift rate of $5 \times 10^{-3}$ m yr$^{-1}$ reported in orogenic settings (Burbank et al., 1996; Beavan et al., 2010). The minimum modelled uplift rate of $1 \times 10^{-5}$ m yr$^{-1}$ was selected because even lower rates led to an impractical computation time required to reach steady state in preliminary model runs. The large parameter space explored in model trials alleviates complications of selecting more limited parameter ranges by bounds that greatly vary globally.

Following uplift in each time step, surface water flow at each node was routed in the single direction of the steepest descent among the eight adjacent nodes. Stream incision and linear diffusion modified elevation further by the Landlab FastscapeEroder and LinearDiffuser components, respectively. Stream power model exponents, $m$ and $n$ were held constant at 0.5 and 1.0, respectively. The factor values for the stream power model coefficient $K$ ranged from $1.0 \times 10^{-6}$ to $1.0 \times 10^{-4}$ yr$^{-1}$. This interval is within reported values of about $2.5 \times 10^{-8}$ yr$^{-1}$ to $2.5 \times 10^{-3}$ yr$^{-1}$ (Stock and Montgomery, 1999; Whipple and Tucker, 1999). The factor values for the hillslope diffusion coefficient $k_d$ ranged from $0.9 \times 10^{-4}$ to $1.0 \times 10^{-1}$ m$^2$ yr$^{-1}$ in a review by Martin (2000). We used a smaller range of $1.0 \times 10^{-3}$ to $1.0 \times 10^{-1}$ m$^2$ yr$^{-1}$.

Stream networks were identified immediately after the initial steady state was reached. Grid nodes were designated as streams if the node contributing drainage area was greater than the value of the sensitivity analysis factor, 'critical drainage area' ($A_c$), that varied between 0.5 km² and 5 km² in the experiment trials. A discrete stream network is defined here as the streams that share an outlet. Outlets existed only at the top or bottom boundary of the model grid in the initial conditions phase. In the

perturb phase described below, networks could temporarily exist in internally drained, endoreic basins with outlets not on a grid boundary.

Each stream network was populated with one species at the end of the initial conditions phase. All species were instantiated with the ZoneSpecies class of SpeciesEvolver. The zone of a species was initially set to the stream network where the species

was populated. Species evolved under the default ZoneSpecies processes, namely dispersal, speciation and extinction (further described in Sect. 3.2.2), meaning custom-made macroevolutionary processes were not used in this study. All species were functionally the same in this experiment, meaning they behaved similarly when presented with the same landscape conditions. Such functional equivalence (neutrality *sensu* Hubbell, 2001) can be set differently in future research. The processes described in this section set the initial conditions of topography, stream networks, and species for the next phase of the experiment.

### 3.2.2 Perturb phase

The steady state topography was perturbed following the final time step in the initial conditions phase and before the first time step in the perturb phase (Fig. 2). The perturbation in a base level fall trial was executed along the bottom boundary of the grid where elevation was decreased by the value of the perturbation magnitude factor, $P_m$. The perturbation in a fault throw trial was executed by a single vertical fault that instantaneously uplifted the right half of the model grid with a throw equal to the

value of $P_m$. The intent of this scenario is to demonstrate drainage reorganisation initiated from a different pattern than base level decline, rather than creating a realistic fault growth model (e.g. Cowie, 1998). $P_m$ spanned values from 0.1 to 100 m. This range falls within observed total fault throw (e.g. Roberts and Michetti, 2004), which is represented by the presence of the fault scarp at model onset. At each time step in the perturb phase, the surface processes were carried out in the same way as in the initial condition phase, using the same factor values for a given trial. The signal of the perturbation through the landscape

was illustrated using

$$\frac{\delta x}{\delta t} = K A^m \tag{6}$$

where $\frac{\delta x}{\delta t}$ is the upstream knickpoint migration rate (Berlin and Anderson, 2007). The maximum $P_m$ value in model trials was within the reconstructed rapid base level fall of 250 m in the Appalachian Mountains (Prince et al., 2011), and observed knickpoint heights, for example the 60 to 110 m range of knickpoint heights on the Roan Plateau (Berlin and Anderson, 2007).

Additionally, main divide migration in each trial was calculated by (1) finding the maximum elevation in each grid column at the first and final time steps of the perturb phase, (2) measuring the distance between the main divide node in the first and final

time steps, and (3) averaging the distance of the main divide nodes to calculate the mean migration of the main divide in the trial.

The macroevolutionary processes (i.e. dispersal, speciation, and extinction) ran subsequent to the surface processes in each time step in this application of SpeciesEvolver (Fig. 2). A schematised version of Fig. 1 is provided in Fig. 3 to demonstrate how drainage reorganisation drove species evolution in the model of this study. In Fig. 3, adjacent stream cells compose a zone of a species. Species dispersal was modelled by resolving the difference in zone extent between an earlier ($T_0$) and later ($T_1$) time step. For example, the zone of stream network 5 (N5) expanded by one cell to the north in $T_1$, thus the species of this zone (E.0) dispersed to this cell between $T_0$ and $T_1$. If a zone of a species was fragmented (due to stream capture, for example), that species divided into one or more child species (clades B and H in Fig. 3). A species became extinct when it was no longer associated with any zones. This occurred when streams in $T_0$ do not overlap any streams in $T_1$ as exemplified by clade D in Fig. 3.

One parameter of the simulated species varied in the trials of the model experiment. This parameter, 'time to allopatric speciation', sets a delay from the time step when the zone of a species fragmented to the time step when speciation is executed by the software. Speciation, when it is triggered by zone fragmentation, is carried out more rapidly as this parameter decreases. The parameter was set to the same value for all species in a trial and it varied from 1 to 100 kyr among the trials, consistent with empirical studies on freshwater fishes (Albert and Carvalho, 2011; Tedesco et al., 2012; Albert et al., 2018), and a theoretical model arising from analyses of molecular phylogenies linking speciation to rare stochastic events that cause reproductive isolation (Venditti et al., 2010; Beaulieu and O'Meara, 2015). For example, if the zone of a species became fragmented and the trial value of this factor was 1 kyr, speciation occurred in the time step following fragmentation because the time step duration of the model is 1 kyr.

The model iterated through time until the time step when topography returned to steady state at which point the trial ended. This final steady state was defined following the same conditions as the initial steady state. Steady state was reached when changes in the mean and standard deviation of elevation over the prior 1 million model years was less than 1 %. In the model designed for this research, evolution effectively ceases following drainage stabilisation, which occurs well before the end of the perturb phase, which is when topography returns to steady state. However, models using SpeciesEvolver in future research can readily incorporate evolutionary processes not exclusively driven by landscape structure, for example sympatric speciation (Lyons et al., 2020). The landscape and biologic model responses in this research were determined from the state of the model immediately following the final time step.

## 3.3 Model response variables

The response variables, which are the model outputs investigated in the sensitivity analyses, were collected from each trial. Topographic relief was the only response collected during the initial conditions phase. It was calculated as the maximum minus the minimum elevation of the grid, excluding the boundary nodes, at the end of the time step when steady state was reached.

Four responses that represent drainage reorganisation and species lineage diversification were collected at the end of the perturb phase. The 'divide percent change response' was calculated by dividing the total cell area of nodes that were drainage divides in either the first *or* the final time step by the total cell area of nodes that were divides in the first *and* final time steps. Divides were identified where there were no upstream nodes (i.e. node drainage area equalled the cell area). The calculation for 'stream percent change response' was similar to divide percent change response. Streams were identified as the nodes with drainage

areas greater than the trial factor value of $A_c$. Divide and stream change response values were used to characterise the percent of grid nodes that changed landform type, and these responses are henceforth collectively referred to as 'landform change'. The 'stream capture count response' is the number of stream captures that occurred during the perturb phase. A stream capture occurred when stream nodes at a time step, $t$ overlapped the stream nodes of another network at $t − 1$. The 'species richness percent change response' was calculated as the percent change of species richness between the first and final time step of the

perturb phase. It was calculated as the final minus initial species count divided by the initial species count.

## 4 Results

The model responses of the 25,600 trials of each scenario are provided in the data repository associated with this paper (Lyons et al., 2019). The video supplement contains animations (V1–V3) of selected trials that exemplify the topographic response to the single base level fall or fault throw perturbation of a trial. At the onset of a trial perturb phase, steepened hillslopes and

stream knickpoints formed where the perturbation originated, which was at base level along the southern model grid boundary or along the fault. Over time, the steepened landscape portion moved away from the perturbation origin, behaving as an erosional wave that locally steepened topography at the wave front and lowered it in its wake. The wave separated the upslope landscape portion yet to adjust to the perturbation from the downslope portion that has adjusted to the perturbation.

The magnitude of the perturbation in the base level fall scenario was a primary control on the migration distance of the main divide and stream knickpoints. The calculation of the main divide migration distance is described in Sect 3.2.2, and its value for each experiment trial is provided in Lyons et al. (2019). The main divide migrated northward by 250 m in trial 5043 with a base level fall of only 2 m (V1). In trials with a similarly small $P_m$ the wave grew and then decayed, all in the southern half of the grid before it reached the main divide. The main divide was driven northward by 7691 m, almost to the northern

boundary, following the 72 m base level fall in trial 12613 (V2). In both of these exemplary trials, streams remain fixed in their course while the wave was in the southern half of the grid. Streams eroded headward once the wave reached the main divide. The wave propagated at the velocity predicted by Eq. (6) (V1–V2). The analytically predicted knickpoint locations in

the supplementary video animations correspond to the location of knickpoints in the modelled landscapes at a given time. In a subset of base level scenario trials (e.g. trial 3639), not one divide or stream node changed during the experiment (Lyons et al., 2019).

The animation of a fault throw scenario exemplary trial demonstrates a different pattern of erosional wave propagation. The wave initiated along the north, west, and south edges of the right block that uplifted instantaneously at the onset of fault throw scenario trials with high $P_m$ relative to the experiment range, including the 72 m throw in trial 12613 (V3 in video supplement). The waves propagated up the watersheds of the right upthrown block until the waves reached the main divide at about the same time. The main divide did not migrate because the base level was the same for the networks that drained to the north and

south boundaries in this scenario. This behaviour is contrary to the base level fall scenario where the main divide migrated towards the upper boundary following the elevation decline only along the lower boundary. Drainage reorganisation was concentrated near the horizontal centre of the grid in the fault throw scenario contrasting with the base level fall scenario where reorganisation was concentrated in the upper 50 % of the grid. The steeper slope across the fault scarp redirected stream flow from the upthrown block to the west that led to drainage capture by stream networks on the downthrown block in 3 % and 56

% of the base level fall and fault throw scenario trials, respectively (Table 2). In a subset of trials, stream segments adjacent to the fault became internally drained before they connected to a network that drained to a grid boundary. Watersheds that did not overlap the fault, or were not immediately adjacent to watersheds that overlapped the fault, did not contain networks that reorganised.

**4.1 Topographic relief and landform change**

Topographic relief was calculated once elevation reached steady state in the initial conditions phase of each trial. Relief varied among the trials with a maximum trial relief of 11,055 m (Table 2). Most trials contained low relief relative to the maximum relief in the experiment (Fig. 4), owing to the distribution of model factor values. Relief was less than 1000 m in 89 % of trials, and relief was greater than 8000 m in only 0.12 % of trials. In Sect. 5.5 we provide considerations of the few model trials with relief greater than observed on present-day Earth.

The total order Sobol indices of $U$ and $K$ were the greatest among the factors, indicating relief was most influenced by $U$ and $K$ (Fig. 5a). $U$ and $K$ individually contributed to about half of the variance of relief as indicated by the first order indices. The other half—represented by the difference between the total and first order indices of these factors—was controlled by second and higher order effects. The only factor pair with a large second order index was $U$ and $K$ (Fig. 5b), indicating that relief in a

given trial was influenced by the interaction of these factors, which is expected because $U$ and $K$ together set relief as specified in Eq. (6). The outcome of this interaction is presented in Fig. 5c. Relief increased with $U$, and for a given value of $U$, relief decreased with an increase in $K$.

$U$, $K$ and $P_m$ were the factors that most influenced divide and stream percent change during the perturb phase (Fig. 6a–d). The divide and stream change model responses, collectively referred to as landform change, indicate the proportion that these landforms relocated during trials as described in Sect. 3.3. Here we compare the landform change responses to steady state relief, rather than compare $U$ and $K$ individually to landform change, because (1) $U$ and $K$ together predict relief, and (2) the relationship between relief, $P_m$ and the landform change responses differed between trials with relief above versus below about 100 m, which coincides with the maximum $P_m$ value. Relief, $P_m$ and landform change increased together in the trials for which relief was less than 100 m (Fig. 7a–b), which was the case in exemplary trial 5043 where $P_m$ was 2.03 m. In these trials, the change in divide and stream locations was most concentrated near the initial position of the main divide in both scenarios, and also near the fault trace in the fault throw scenario (e.g., trial 5043; Fig. 7a–d). Stream tips contracted or expanded without capturing segments from adjacent networks (Fig. 8b,d). As $P_m$ increased, for example in exemplary trial 12613 where $P_m$ was 72 m and relief was also less than 100 m, the relocation of divides and streams extended to a greater portion of the model grid (Fig. 8e–h).

The change in the position of streams and divides in the base level fall scenario was concentrated near the initial position of the main divide in the trials for which divides and streams were mobile. In the trials where $P_m$ was greater than relief, streams and divides relocated throughout the northern half of the grid as the main divide drove further northward (e.g. trial 12613; Fig. 8e–f; V2 in video supplement). South-flowing streams extended almost to the northern boundary and tended to reoccupy channels initially incised by north flowing streams (Fig. 8f). Up to about 80 % stream nodes were changed when relief was less than 100 m in this scenario (Fig. 7e).

Landform change in the fault throw scenario was concentrated near the fault trace. The percentage of divides that changed location during the perturb phase reached only about 30 % when relief was less than 100 m except in the few trials where both (1) $k_d$ was near the experiment maximum of $10^{-1}$ m$^2$ yr$^{-1}$ and (2) relief approached 100 m (Fig. 7b). Maximum landform change was lower in this scenario because topography was primarily perturbed in catchments near the fault compared to the base level fall scenario where a greater proportion of landforms changed in the wake of the erosional wave that spanned the width of the grid. For this reason, the $P_m$ total order index of divide change was relatively lower in the fault throw scenario (Fig. 6a–b). In trials with a relatively large $P_m$, for example the 72 m fault slip in exemplary trial 12613 compared with the 2.03 m slip in trial 5043, divide and stream relocation was concentrated around the fault (Fig. 8g–h), and the influence of $k_d$ on divide change became relatively more influential than strictly $P_m$ as described below. In both scenarios, a greater $P_m$ produced a steeper erosion front that propagated further and disrupted drainages in its passage. The relatively higher second order Sobol index of factor pair $K$ and $P_m$ in most of the landform change responses (Fig. 9a,c,d) indicates the relative importance of the interactions among these factors.

Divide change increased with $k_d$ when relief was greater than 100 m in both scenarios (Fig. 7c–d). The increase of $k_d$ with divide change at greater relief, combined with the low range of divide change at low relief in the fault throw scenario, elevated the importance of $k_d$ to this response in this scenario (Fig. 6a–b). In both scenarios, divide change reached about 40 % in trials where relief was near 100 m and $k_d$ was near the experiment maximum of $10^{-1}$ m$^2$ yr$^{-1}$ (Fig. 7c–d). In these trials, the stream

networks and area of catchments tended to not change substantially, although many divides shifted less than 500 m (e.g. trial 21395; Fig. 8i,k). Trial 21395 is within the area in Fig. 7c–d where $k_d$ increased with divide change. This area corresponds to the trials where $K$ is less than $2 \times 10^{-6}$ yr$^{-1}$, the values nearest to the experiment minimum of this factor.

The relative influence of the factors on stream change was similar to divide change with a few exceptions (Fig. 6a–d). The

total effect of the initial elevation seed was relatively greater for stream change in the fault throw scenario. The total order effect of $K$ was lower for stream change than divide change in the fault throw scenario. Although streams changed in response to the combined values of multiple factors (Fig. 9d), mostly along with $K$. The total effect of $k_d$ for stream change was also lower in both scenarios. Stream change was minimally affected by $k_d$ because diffusion minimally affects channels (Fig. 8j,l).

## 4.2 Controls on stream capture occurrence

The frequency and grid location of stream captures differed between the two scenarios. Captures occurred in 3 % and 56 % of the trials in the base level fall and fault throw scenarios, respectively (Table 2). Captures in the trials of the base level fall scenario tended to be located in one of two grid areas. Near the main divide once the erosional wave reached this divide, a stream of a southern network captured a segment of a northern network as the erosional wave drove northward expansion of the southern networks (V2 in video supplement). Captures in this scenario also tended to be located near the lower boundary

when nearby streams were diverted to different outlets following base level fall (e.g. trial 12126; V4).

Streams were captured across the fault trace in the fault throw scenario. In many trials, closed basins (i.e. endorheic) were formed along the fault and were involved in stream capture. First, stream segments detached from the initial networks where the instantaneous fault slip formed a scarp that blocked streamflow and formed closed basins (V3). Over time these basins and

the stream segments within them continued to uplift and erode as the local relief declined. The detached segment within the closed basin was captured by a stream that breached the closed basin, and hillslopes within the basin were soon dissected again. In few trials, captures also occurred where the upper stream reaches of networks on the upthrown block were captured by a network on the downthrown block.

The initial elevation seed factor had the greatest total order effect and interacted with many factors to influence capture occurrence in the fault throw scenario (Fig. 6f; Fig. 9f). Stream networks emerged during the initial conditions phase from the randomly generated elevation noise at the onset of a model trial. The noise was set by the value of the seed that led to the initial

stream networks. The initial location of stream networks was important only in the fault throw scenario because only the networks near the fault were perturbed.

Multiple other factors contributed to the number of captures in the trials of both scenarios (Fig. 6e–f). Factors $U$, $K$, $P_m$ and $A_c$ were similarly important within a given scenario. Confidence intervals of factors were large in the base level fall scenario where captures occurred in relatively few trials. Nevertheless, the interaction of $P_m$ and $K$ was elevated above other interactions in this scenario (Fig. 9e). We examined capture count versus the ratio of $P_m$ and relief, as the result of $U$ and $K$, given the control that these factors acted together to influence landform change. Streams more readily changed location and the number of captures increased rapidly in the trials for which $P_m$:relief exceeded 1 (Fig. 10a–b). In trials well below this value, captures were fewer and stream change was limited to minor expansion and contraction of stream tips (e.g., trial 5043; Fig. 8b,d). Multiple captures did occur when $P_m$:relief was slightly less than 1 in numerous trials of the fault throw scenario (Fig. 10b,d). The stream networks fragmented in these trials, forming endorheic basins that existed for a few time steps, and then the network segments reconnected to a configuration similar to the pre-perturbation configuration. This reorganisation sequence incremented capture count as the fragmented network segments reintegrated.

$A_c$ contributed to the variation in capture count among the trials (Fig. 6e–f). Capture count increased with decreasing $A_c$ (Fig. 10c–d). This relationship is most apparent where $P_m$:relief is near 1 because this ratio value was also required for capture count to increase. Few captures occurred even when $A_c$ was near the experiment minimum of this factor in trials that $P_m$:relief was well below 1.

**4.3 Controls on species richness**

The relationships among relief, $P_m$, and species richness change differed between the scenarios. Species richness increased in 0.2 % and 39.4 % in trials of the base level fall and fault throw scenario, respectively (Table 2). Species richness did not change or decreased in the majority of the base level fall trials (Table 2; Fig. 11a). A decrease in richness occurred when the final species count was less than the initial count, meaning extinction was more common than speciation. Extinction in this simple implementation of SpeciesEvolver occurred only when all of the stream networks of a species disappeared. A network disappeared when its minimum drainage area decreased below $A_c$. Species richness decreased up to 78 % when topographic relief was less than 100 m in a trial (Fig. 11a). Below about 100 m relief, increasingly greater $P_m$ was required for a loss in species richness. In the fault throw scenario, a greater increase in species richness occurred in a subset of trials with low relief and even moderate $P_m$.

Stream capture count and species richness increased together with wide variability (Fig. 11c–d). Trials in the base level fall scenario with relatively little time to allopatric speciation increased with capture count and species richness change. Overall

the relationship of time to allopatric speciation with capture count and species richness change is unclear given the relatively few trials with captures in this scenario. In the fault throw scenario, species richness increased as the time to allopatric speciation decreased for a given capture count.

5     The relative influence of factors on species richness differed between the scenarios more than the other responses (Fig. 6). $U$, $K$, and $P_m$ were the factors with the greatest total order indices of species richness percent change in the base level fall scenario. Additionally, the relative magnitudes of the species richness change Sobol indices were more similar to the landform change responses than capture count in this scenario (Fig. 6g; Fig. 9g). The relative importance of $P_m$ to species richness change was comparably lower in the fault throw scenario where the initial elevation seed and $k_d$ total effect indices were comparably 10   greater. The relative magnitudes of species richness change Sobol indices were more similar to capture count than landform change responses in the fault throw scenario (Fig. 6h; Fig. 9h).

The time frame of speciation following a perturbation differed among the trials. This is exemplified in the trials animated in the video supplement and the phylogeny of their simulated species (Fig. 12). Speciations and extinctions ceased soon after the 15   perturbation in exemplary trial 12126 of both scenarios as well as trial 12613 of the fault throw scenario. Speciations and extinctions continued to near the end of trial 12613 in the base level fall scenario where captures did not occur until the erosional wave reached the main divide (V5 in video supplement). The lineage of clade F in trial 12613 of the fault throw scenario became most diverse with 4 species where two stream networks were captured by a third network soon after the perturbation (V6). Clade D in both scenarios of trial 12613 went extinct in the time step following the perturbation.

20   **5 Discussion**

**5.1 Are landscapes with low or high topographic relief more susceptible to drainage reorganisation?**

The ratio of the relative value of trial $P_m$ to steady state relief was a primary control on the degree of drainage reorganisation. The extensiveness of drainage reorganisation increased with this ratio. In model trials, an erosional wave was initiated by a vertical magnitude equal to the trial value of $P_m$, and the magnitude of the wave tended to decay as it approached divides. 25   Cross-divide difference in relief, an indicator of divide instability (Whipple et al., 2017), at the main divide seemingly remained near zero when $P_m$ was small relative to initial relief, thus divides did not migrate. The difference of cross-divide relief increased if the wave did not fully decay before reaching the divide. Divides migrated and streams shifted more often as the trial ratio of $P_m$ and initial relief approached and exceeded unity. The divide continued to migrate until the erosional wave decayed or the main divide reached a grid boundary. The magnitude of past perturbations and the spatiotemporal decay of their 30   waves are difficult to determine in real landscapes, although relief and other topographic metrics can be measured (Wobus et al., 2006; Willet et al. 2014; Whipple et al., 2017; Guerit et al., 2018). Model results imply that real world regions or landscape

portions with low relief are especially susceptible to extensive divide migration and stream capture compared to areas with greater relief in response to a perturbation of a similar magnitude, when all else is relatively less effective at stabilising the organisation of drainages.

## 5.2 What process parameters influence drainage reorganisation susceptibility for landscapes with a given relief?

The factors, $P_m$, $K$, and $U$ exerted the greatest influence on drainage reorganisation out of all experiment factors. Drainage reorganisation increased with $P_m$ for a given relief (Fig. 7). In trials of the base level fall scenario, the migration of the main divide increased with $P_m$, which also increased the opportunity of cross-divide stream tips to capture. In trials of the fault throw scenario, reorganisation increased with $P_m$ because greater slope changes across the fault more likely redirected flow. Captures in this scenario often occurred as stream segments in internally-drained basins were reintegrated into stream networks draining
to grid boundaries. This sequence of processes occurred similarly in the Apennines where streams captured intermontane internally-draining basins formed by subsidence related to normal faulting (D'Agostino et al., 2001), although in the model without the broad topographic bulge induced by mantle upwelling as is the case beneath the Apennines. In both scenarios, $U$ and $K$ strongly influenced landform change and capture count, but that is because these factors set relief. $K$ had a greater influence than $U$ on most landform responses because $K$ also set erosional wave celerity. Erosional waves can propagate
further when rock erodibility is greater, leading to greater change in the location of divides and streams as well as more stream captures. High erodibility can also correspond with low relief landscapes, increasing the susceptibility of drainage reorganisation following perturbations. Few real-world landscapes have homogenous erodibility at relatively large scale and few modelling efforts have investigated the dynamics of heterogeneous erodibility (e.g. Forte et al., 2016), which likely affects drainage reorganisation as well as macroevolutionary processes. Overall, drainage reorganisation in the model shared
similarities with real world examples, e.g. Seagren and Schoenbohm (2019) who concluded that uplift history, erodibility, and local base level controlled the pattern of drainage reorganisation in their study landscape in northwest Argentina.

The controls on drainage reorganisation responses transitioned where trial steady state relief is about the maximum value of $P_m$, which was 100 m (Fig. 7). Stream location change remained less than 30 % in the trials with relief greater than 100 m
because the experiment maximum $P_m$ was 100 m. In these trials stream topology before and after the perturbation were similar because the erosional wave decayed before it reached the main divide. In the trials with relief below 100 m, stream location change reached about 80 % because the erosional wave could reach or pass the initial position of the main divide, for the finer subset of trials in which $P_m$ was near or exceeded the magnitude of trial relief. Also in the trials with relief below 100 m, a greater $P_m$ was required to elicit a given stream or divide location change as relief increased because $K$ decreased with relief
and erosional waves travelled shorter distances as $K$ decreased. Stream capture was also more prevalent when relief was below 100 m, or stated more directly, when the ratio of $P_m$ to relief was near or above one (Fig. 10), as flow direction more readily shifted where the perturbation could alter the existing relief structure. A greater proportion of divides changed location in the

trials where steady state relief exceeded $P_m$, $k_d$ was near the experiment maximum factor value of $10^{-1}$ m$^2$ yr$^{-1}$, and relief was greater than 100 m (Fig. 7). Greater diffusion produces lower local relief on either side of a divide, although divides moved minimally under this combination of factors (e.g. trial 21395; Fig. 8i,k). Conversely, divides migrated further distances, streams relocated more extensively, and captures were more frequent in the trials in which initial relief was relatively low (e.g. trial 12613; Fig. 8e–h).

Other factors modulate drainage reorganisation under certain combinations of factor values and conditions set by the scenarios. The initial elevation seed, which influenced the locations of the initial streams, was indicated by a sensitivity analysis to be the most important factor studied in this research to stream change and capture occurrence in the fault throw scenario. The influence of the seed value and stream locations on drainage reorganisation would decrease relative to the other model factors in a landscape with multiple faults because more streams would more likely be near a fault. $A_c$ was more important to stream capture occurrence in the base level fall scenario than the fault throw scenario. This factor effectively set the distance between streams of adjacent networks. Capture occurrence increased as $A_c$ decreased because shorter divide migration distance is required to result in a capture in the model. However, overlap of stream tips across migrating divides in successive time steps of our model might not have been designated as captures if the time step was shorter and instead would simply be migrating divides.

**5.3 Do the same parameters that lead to drainage reorganisation also control riverine species diversity within a landscape?**

Base level fall and fault throw altered species richness differently. Overall, species richness increased due to the conditions that led to more captures, and richness decreased due to the conditions that led to stream network disappearance. In the base level fall scenario, species richness most often decreased during trials because extinctions were numerous and captures occurred in few trials. Extinctions following stream network disappearance most often occurred in this scenario as the main divide reached near the upper boundary that decreased the drainage area of the catchments to the north of the divide. The factors that drove the main divide and controlled the celerity and magnitude of the erosional wave were $P_m$ and $K$, which were dominant factors in controlling species richness in the base level fall scenario. The number of species typically decreased between the start and end of a base level scenario trial. Species richness decreased in 39.6 % of the trials in this scenario (Table 2). This explains why the combination of factors with high Sobol indices were more similar between species richness and landform change rather than stream capture count. Few captures and associated speciation events occurred in this scenario. Extinction related to divide migration was more common.

In the fault throw scenario, the combination of factors with high Sobol indices were most similar between species richness and stream capture count. Following a capture, the inhabitant species are located in multiple zones and this triggers a speciation event following the delay set by the time to allopatric speciation parameter, meaning that a species was gained in each

additional zone where the parent species dispersed (V6 in video supplement). Therefore, species richness and capture count should increase together, which was demonstrated especially in the fault throw scenario. Species richness increased in the majority of trials of this scenario, especially those with sufficiently large $P_m$ and low relief (Table 2; Fig. 11b). Fault slip detached stream segments from the initial networks, which triggered speciation because the zone of species became fragmented

(V6). In the following time steps these segments were captured by a stream network, and the new species dispersed across a greater area. As species richness increased in a network, the number of new species associated with a capture increased.

Other factors were important to species richness beyond the factors important to stream capture discussed in Sect. 5.2. The influence of the time to allopatric speciation parameter was evident only in the fault throw scenario (Fig. 6h), because this

factor was relevant only to speciation that was common in this scenario. Species richness increased the most when speciation time was relatively short (Fig. 11d). Fewer species were spawned when wait time was long because stream segments would reconnect in the trials where captures were limited to temporary fragmentations of stream networks that did not move. Time to speciation represents the speed at which species evolve. Rapidly occurring stream captures lead to greater species richness if species evolve faster. Slowly evolving species will not speciate for the captures that temporarily disconnect a stream segment

from its original network.

Speciation events were more frequent when $A_c$ was relatively small because streams extended nearer to divides, effectively reducing the perturbation magnitude required for a stream capture. Extinctions in the experiment model will be more frequent when $A_c$ is relatively large because smaller drainages along boundaries are more susceptible to shrinking below $A_c$, which

then causes the network to disappear and its species to become extinct. Additionally, fewer of the closed basins that form in some trials immediately after fault slip will contain networks when $A_c$ is relatively large because the closed basins smaller than $A_c$ do not contain drainage area great enough to contain streams.

This application of SpeciesEvolver began with one species per stream network to investigate lineage development following

a single perturbation. We hypothesise that multiple perturbations will tend to push the number of species and the areal extent that the species inhabit towards the widely reported power-law relationship between these factors (He and Hubbell, 2011). As we brought the modelled landscape to steady state, the initial conditions in future applications of SpeciesEvolver can begin by populating the landscape with a power-law relationship of species set by range area, depending upon the intent of the model. We strictly used a modelling approach in this study to demonstrate a framework in which landscape and life evolution can be

investigated together. SpeciesEvolver is capable of site- and taxon-specific studies, including running with a digital elevation models of a real landscape. Linking models, real landscapes and life can be aided by geomorphic abiotic parameters, such as elevational landscape connectivity (*sensu* Salles et al., 2019).

## 5.4 Numerical test of the river capture hypothesis of riverine species diversification

The river capture hypothesis is built on the observation that the diversity of many terrestrial organismal groups are greatest in mid to high relief landscapes while fish are often most diverse in lowland basins (e.g. Hoeinghaus et al., 2004; Grenyer et al., 2006). Badgley (2010) proposed that this phenomenon can be explained by the lower extinction rates of fish in tectonically passive regions, which often have low relief compared to tectonically active regions. Low extinction rates and high lowland fish diversity can be further explained mechanistically through stream captures that both increase diversity through speciation and reduce the probability of extinction as species disperse across a larger area (Albert et al., 2018). The greatest diversification of species lineages in our model-based experiment was in low relief landscapes. This result is consistent with the river capture hypothesis (*sensu* Albert et al., 2018), illustrating a mechanism of riverine species diversification in low relief landscapes where drainage reorganisation was most extensive in our modelling. Further studies that couple empirical data analyses with modelling of landscape and biological evolution are needed to further investigate this hypothesis.

## 5.5 Limitations of model experiment and future directions

The model experiment was designed to simulate drainage reorganisation carried out by the processes most commonly implemented in fundamental landscape evolution models, namely stream power incision and hillslope diffusion. Limitations potentially critical to drainage reorganization include the limitations of the stream power model in general described in Lague (2014). All discharge in our model effectively contributes to incision throughout the grid at each time step. Implementing the stream power model with a stochastic incision threshold including discharge variability provides an avenue to improve predictions of both steady state topographic structure and knickpoint propagation (Lague, 2014), both of which directly impact drainage reorganisation. Divide migration can progress by hillslope processes beyond diffusion, i.e. mass wasting (Dahlquist et al., 2018). As a consequence, divide migration may have been underpredicted especially in the relatively few trials with high relief (Fig. 4). Speciation in our model was driven by stream capture and not drainage migration. A portion of high magnitude and low frequency landslides may have produced captures of headwater stream tips. Lithologic erodibility varied from one model trial to the next. Uniform erodibility in space and time within a trial is another limitation of this research. Spatiotemporal variation in erodibility can lead to drainage reorganisation where reorganisation may not have occurred without erodibility variation (Forte et al., 2016; Gallen et al., 2018; Harel et al., 2019). Existing Landlab capabilities offer opportunities to begin addressing some limitations in our model, including simultaneously transporting fluvial sediment and eroding bedrock (Shobe et al., 2017), emplacing lithologic heterogeneity in the model grid (Barnhart et al., 2018), and identifying areas with elevated probability of landsliding (Strauch et al., 2018).

Terrestrial topographic relief not observed on Earth was constructed during the initial conditions phase in a small portion of trials (e.g. 0.12 % of trials with relief greater than 8000 m; Fig. 4), despite using reasonable values for uplift rate and lithological erodibility. We primarily attribute this to no mass wasting in the model and a fixed rainfall rate of 1 m yr$^{-1}$. We do not anticipate

that our interpretations throughout Sect. 5 and conclusions in Sect. 6 are affected by the unrealistically high relief in the few trials. Our results regarding relief are primarily important relative to the magnitude of the simulated perturbation, which reached up to 100 m. The relief of the majority of trials (89 %) was less than 1000 m owing to the factor value sampling procedure described in Sect. 3.1.

The macroevolutionary processes of the simulated riverine species were designed to not obscure potential links between drainage reorganization and species richness model responses. Model trials began with a single species per network and additional species could join the single initial network species through stream capture. The model did not include predation, competition for resources, or within stream networks limits on species range, some of which are included in spatially explicit ecological models (e.g. Rangel et al., 2018). Spatially variable biodiversity emerges throughout individual real stream networks, for example the often observed downstream increase in diversity (Grossman et al., 2010). Future studies can adapt SpeciesEvolver taxon objects (e.g. species) to interact with other objects and set their dispersal ability to only portions of a stream network (e.g. by stream order). The dynamics of dispersal strongly impact gene flow and macroevolution (Coyne, 1992). Knickpoints that exceed the ability of upstream passage of riverine organisms can impede or block gene flow of intraspecies populations (Crispo et al., 2006). The dispersal of SpeciesEvolver taxa can be restricted across steep stream reaches to investigate the impact of knickpoints on riverine species. Overall, developing techniques to compare empirical and model data is of utmost importance in future research. This challenge was not examined in our strictly model-based study as simulated species richness was the biological variable examined. One approach is explicitly modelling the genetics of individual organisms that will enable comparison of empirical and modelled datasets using population genetics statistics.

**6 Conclusions**

We first investigated the conditions in which the drainage networks of a landscape evolution model reorganise. Sensitivity analyses indicate multiple factors influence the occurrence and expansiveness of drainage reorganisation. Reorganisation was extensive when the magnitude of the topographic perturbation exceeded that of the initial relief. The erodibility coefficient of the stream power model was exceptionally important to drainage reorganisation because it controlled both topographic relief and the celerity of the erosional wave that propagated through the landscape following the perturbation. Secondarily, the number of stream captures in a trial was influenced by the critical drainage area of stream initiation and by the initial stream topology when the perturbation was carried out by the throw of a fault. The complexity of these results yielded by a simple model with few parameters helps to demonstrate why the real-world behaviour of drainage reorganisation is elusive.

We also investigated the dynamics of riverine species diversification following drainage reorganisation in landscapes with low and high relief. To accomplish this, we used a new model component that simulates macroevolutionary processes coupled with surface processes. This component was used in the same model trials of the drainage reorganisation sensitivity analyses. Trial

species richness increased by up to 518 % even though each trial was subjected to only one topographic perturbation, although with simplified extinction and no interspecies dynamics. The model results illustrate how a landscape with few species can evolve into a biodiversity hotspot following drainage reorganisation, at least for some period following a perturbation. The results also illustrate how the lineages of riverine species can diversify as a consequence of extensive drainage reorganisation especially in low relief landscapes—contrary to the diversity of other terrestrial organismal groups in high relief landscapes—all supporting the river capture hypothesis posed in prior research.

Drainage reorganisation is difficult to document and few direct observations exist (e.g. Stokes et al., 2018), in part because evidence of reorganisation is minimally preserved as drainages continuously adjust to boundary conditions. Landform preservation, disparate timescales of the aforementioned landscape evolution process components, and the formation of species-dense assemblages of riverine organisms are but a few of the challenges to relate the evolution of a landscape with its lineages. Future applications of the SpeciesEvolver modelling tool can further explore the mechanisms by which organismal lineages respond to landscape changes, and provide opportunities to pursue taxon-specific and region-specific questions regarding the interactions between aquatic biotas and their environments. The SpeciesEvolver component in Landlab (Lyons et al., 2020) is a contribution to the arsenal needed to untangle the topographic controls on biodiversity, and this insight may lead to our ability to learn about landscapes from the species that inhabit them.

**Data Availability**

Sensitivity analysis trial factor values, model responses, and Sobol indices are available at https://doi.org/10.5281/zenodo.3893629 (Lyons et al., 2019).

**Video Supplement**

Videos are animations of model output of selected trials that exemplify aspects of drainage reorganization and lineage diversification. All videos animate topographic slope of the model grid on the left. The following videos animate a selected longitudinal channel profile to the right of the grid: V1 (https://doi.org/10.5446/43655), V2 (https://doi.org/10.5446/43656), and V3 (https://doi.org/10.5446/43657). The following videos animate a plot of capture count and species richness to the right of the grid: V4 (https://doi.org/10.5446/43658), V5 (https://doi.org/10.5446/43659), and V6 (https://doi.org/10.5446/43660). Animations begin at the final time step of the initial conditions phase immediately prior to perturbation ('elapsed time' is 0 yrs in the animations) and continue until the end of the perturb phase. Meaning, the first animation frame depicts topography at the initial steady state and the final frame is the second and final steady state.

## Author Contribution

N. J. Lyons designed and developed SpeciesEvolver, conducted the experiments, and wrote the manuscript. N. J. Lyons and P. Val devised the model scenarios. P. Val, J. S. Albert, J. K. Willenbring, and N. M. Gasparini reviewed the manuscript.

## Competing Interests

5   The authors declare that they have no conflict of interest.

## Acknowledgements

Support for this project was provided by the Tulane University Oliver Fund Scholar Award and NSF OAC Award 1450338. Thorough reviews by Laure Guerit, an anonymous referee, and editors Sébastien Castelltort and Josh West greatly improved the manuscript. General software development support was provided by the NSF funded CSDMS project. High performance 10   computing resources were provided by Technology Services at Tulane University, New Orleans, LA.

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

**Table 1.** Parameters of model trials.

| Constant | |
|---|---|
| time step | 1000 yr |
| drainage area exponent, $m$ | 0.5 |
| channel slope exponent, $n$ | 1.0 |

| Sensitivity analysis factor | |
|---|---|
| initial topography seed | $1 - 20,000$ |
| uplift rate, $U$ | $10^{-5} - 10^{-3}$ m yr$^{-1}$ |
| erodibility coefficient, $K$ | $10^{-6} - 10^{-4}$ yr$^{-1}$ |
| diffusion coefficient, $k_d$ | $10^{-3} - 10^{-1}$ m$^2$ yr$^{-1}$ |
| critical drainage area, $A_c$ | $5 \times 10^5 - 5 \times 10^6$ m$^2$ |
| perturbation magnitude, $P_m$ | $10^{-1} - 10^2$ m |
| time to allopatric speciation, TAS | $10^3 - 10^5$ yr |

**Table 2. Response summary statistics.** The perturb phase statistics are calculated separately for the trials when a given response, $R$ was less than, equal to, or greater than 0. Mean values of $R$ were calculated for the trials where $R$ was not equal to 0. The plus-minus values associated with the mean $R$ values provide the standard deviation of change for all model trials of a scenario where $R$ was not equal to 0.

| Response, $R$ | Statistic | Initial conditions phase | |
|---|---|---|---|
| Topographic relief at steady state | minimum | | 0.9 m |
| | mean | | 447 m |
| | maximum | | 11,055 m |
| | Perturb phase: | Base level fall | Fault throw |
| Divide percent change | trial count: $R = 0$ | 265 (1 %) | 173 (1 %) |
| | trial count: $R > 0$ | 25,335 (99 %) | 25,427 (99 %) |
| | mean $R$: $R > 0$ % change | 14.85 ± 13.88 % | 11.66 ± 10.12 % |
| Stream percent change | trial count: $R = 0$ | 1405 (5 %) | 1214 (5 %) |
| | trial count: $R > 0$ | 24,195 (95 %) | 24,386 (95 %) |
| | mean $R$: $R > 0$ % change | 17.99 ± 23.51 % | 8.55 ± 7.94 % |
| Capture count | trial count: $R = 0$ | 24,919 (97 %) | 11,272 (44 %) |
| | trial count: $R > 0$ | 681 (3 %) | 14,328 (56 %) |
| | mean $R$: $R > 0$ captures | 2.35 ± 2.11 | 2.44 ± 2.14 |
| Species richness percent change | trial count: $R < 0$ | 10,135 (39.6 %) | 5380 (21.0 %) |
| | trial count: $R = 0$ | 15,412 (60.2 %) | 10,140 (39.6 %) |
| | trial count: $R > 0$ | 53 (0.2 %) | 10,080 (39.4 %) |
| | $R$ mean: $R < 0$ % change | -25.06 ± 19.67 % | -12.27 ± 5.61 % |
| | $R$ mean: $R > 0$ % change | 10.93 ± 4.56 % | 21.73 ± 21.59 % |

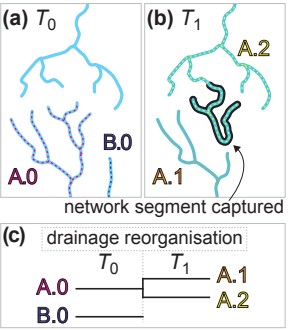

**Figure 1. Conceptual model of drainage reorganisation and riverine species macroevolution.** Three stream networks exist in a hypothetical landscape at the time, $T_0$ (a). Riverine species, A.0 inhabits the lower-left stream network and B.0 inhabits the lower-right network. Drainages reorganised between $T_0$ and a later time, $T_1$. Reorganisation was carried out by a stream capture where a network segment broke off the lower-left network and joined the upper network (b). Members of species A.0 that existed in the captured segment dispersed throughout the upper network creating two populations of this species in distinct stream networks that speciated child species, A.1 and A.2. Drainage reorganisation also led to the stream network of B.0 to disappear, driving the extinction of this species. The lineage history of the species before and after drainage reorganisation is presented in a phylogenetic tree (c). After Albert et al. (2011).

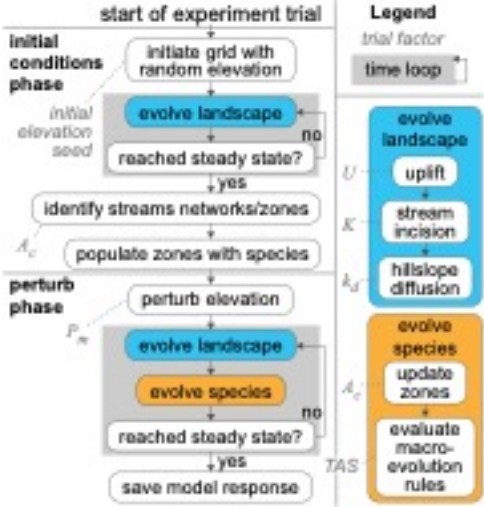

**Figure 2. Progression of an experiment trial.** The base level fall and fault throw scenario trials progressed as outlined in this flow chart. The two phases of the model both included a time loop. The steps in the time loop were repeated until topography reached steady state. The evolution processes in the time loops are detailed on the right. Dashed lines connect trial factors to the steps that the factors parameterise.

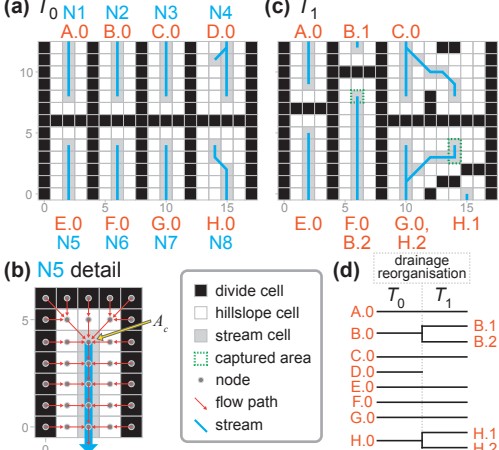

**Figure 3. Downscaled schematic of modelling approach.** (a) A schematised steady-state landscape where the main divide separates 8 stream networks (N1…N8) that each flow to either the north or south boundary. (b) The species and zones of SpeciesEvolver are defined at the nodes of a Landlab grid. In this study, nodes with a drainage area greater than $A_c$ define the zone of a species. (c) The landscape following reorganisation. N6 and N7 captured areas from adjacent networks. While N3 did extend into the watershed of N4, it did not overlap the stream nodes of the prior time step, therefore N3 did not capture N4 following the strict definition of capture in this study. N4 disappeared because all nodes in the northeast watershed have a drainage area below the critical drainage area. (d) The phylogenetic tree of the species in (a) and (c).

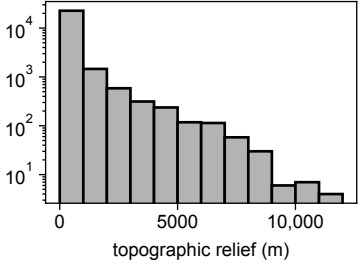

**Figure 4. Histogram of topographic relief.** The plotted data is the topographic relief at the trial end of the initial conditions phase of each trial. Note the y-axis is logarithmic.

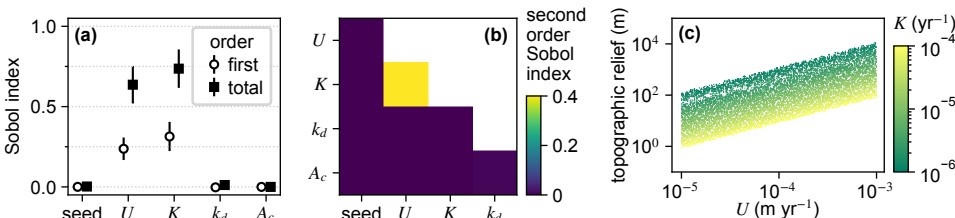

**Figure 5. Sobol indices of topographic relief**. (a) The first and total order Sobol indices of relief at the initial steady state. Model input factors are on the x-axis where seed is the initial elevation seed. (b) Second order Sobol indices of relief. Factors are on the x- and y-axes. (c) Relief versus $U$ and $K$. Each point represents one of the unique steady state landscapes created in the initial conditions phase.

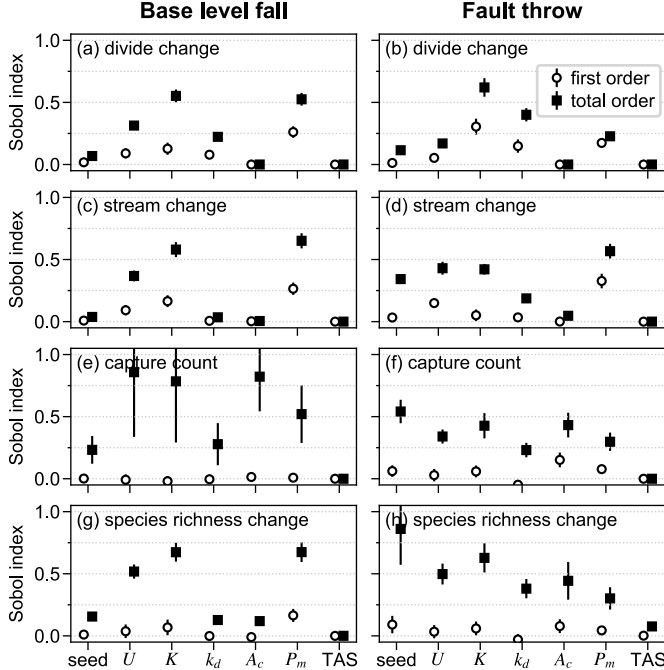

**Figure 6. First and total order Sobol indices of drainage reorganisation responses.** The factors are along the x-axis for each of the responses (a–h) where seed refers to the initial elevation seed.

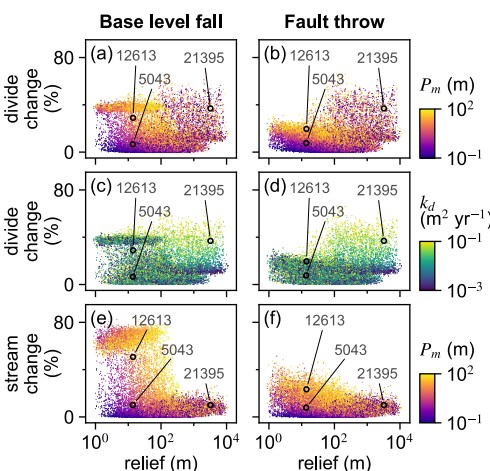

**Figure 7. Landform change responses versus initial relief.** Responses of all trials for divide percent change (a–d) and stream percent change (e–f). The labelled points are the IDs of the exemplary trials depicted in Fig. 8 and described in the text.

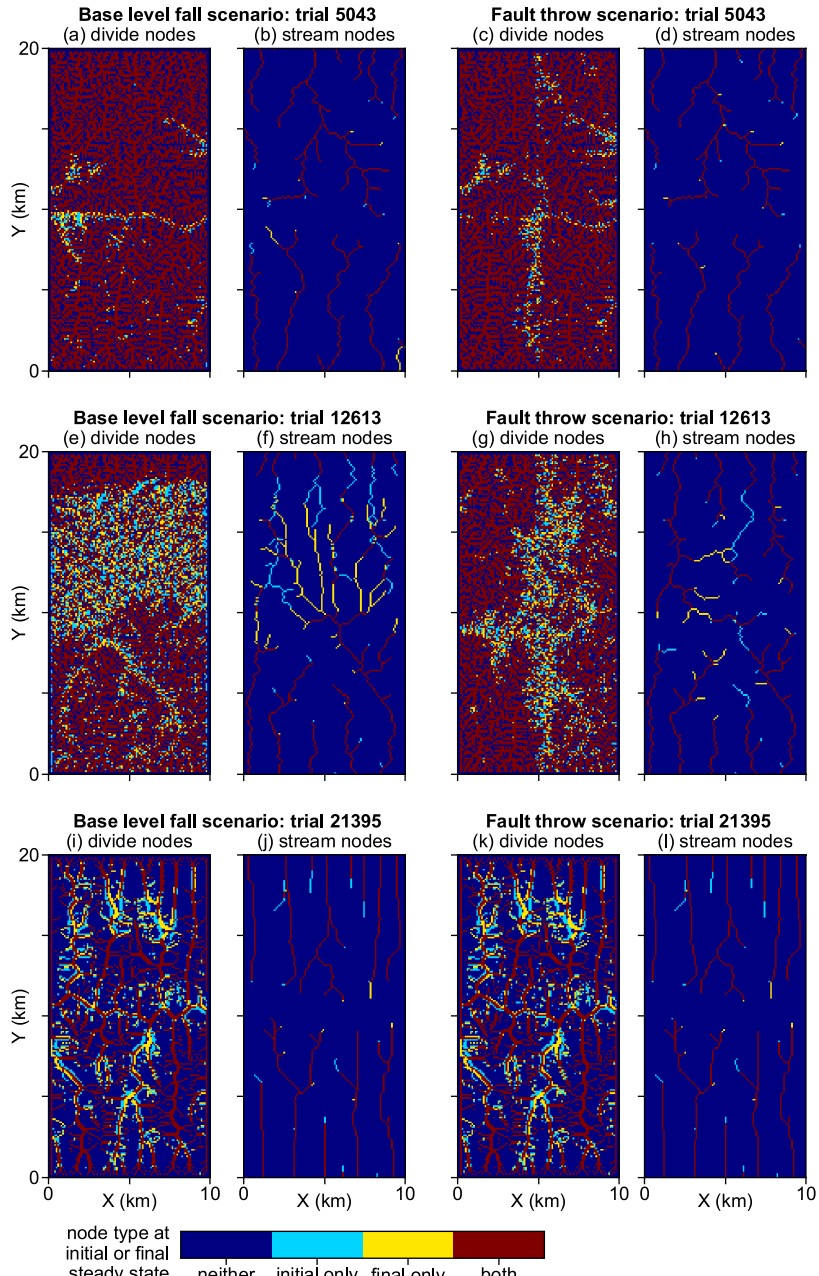

**Figure 8. Landform change of exemplary trials.** This figure illustrates landform change model responses of the trials discussed in Sect. 4 and labelled in Fig. 7. The colour of grid cells symbolises landform type at the initial and final steady state in the model. Blue areas were not the landform type (divide or stream) in a given subplot at the times of either steady state. Red areas were the subplot landform type in both steady state times. Cyan and yellow areas were the subplot landform type in the initial and final steady state, respectively. The parameter values of these trials and all other trials are provided in Lyons et al. (2019).

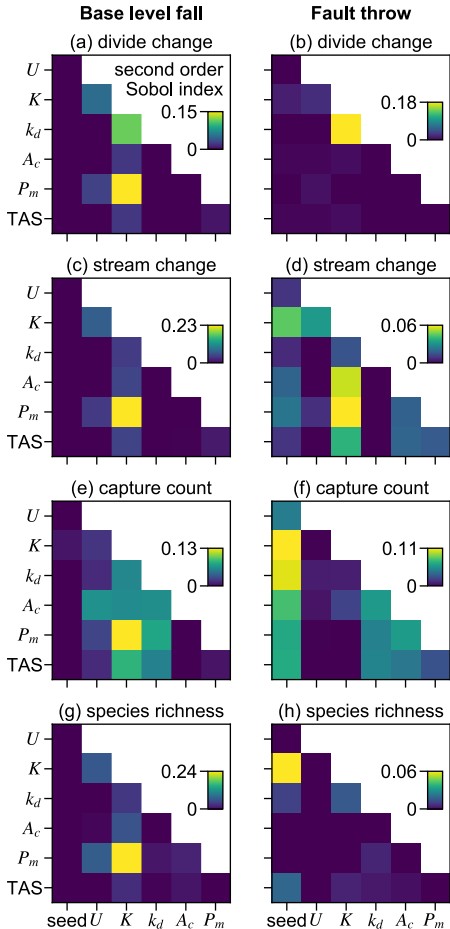

**Figure 9. Second order Sobol indices.** Second order indices of paired model factors for the perturb phase responses. A relatively large value in a subplot indicates that the interaction of the factor pair affects the response more than other factor pairs with lower index values.

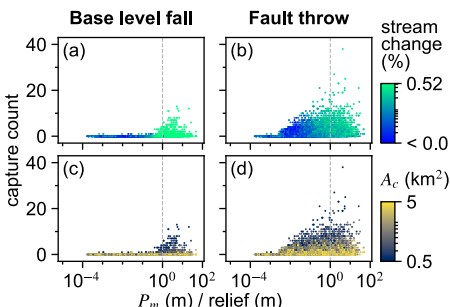

**Figure 10. Capture count versus the ratio of $P_m$ and relief.** $P_m$ and relief are equal at the vertical dashed line.

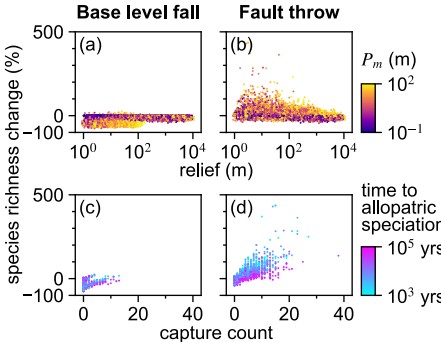

**Figure 11. Species richness percent change.** Species richness change versus relief (a–b) and capture count (c–d).

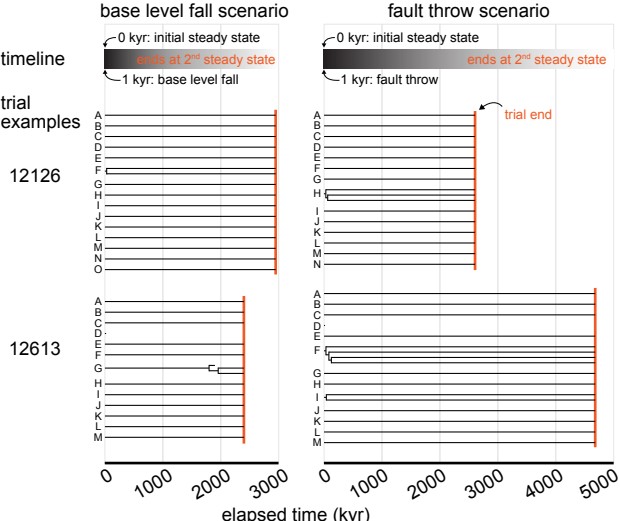

5     **Figure 12. Phylogeny of exemplary trials.** Topography was perturbed by base level fall of fault throw at 1 kyr elapsed since the first steady state was reached. Most of the trials animated in the video supplement are shown. Trial 5043 is not included. Species did not change in this trial because no stream networks disappeared or were captured. The lineages of clades in a trial are labelled alphabetically. Speciation events occurred where lineages split and extinctions occurred where lineages terminated before the end of the trial.