# Peer review of "Topographic controls on divide migration, stream capture, and diversification in riverine life"

_Earth Surface Dynamics, 2019_

## Referee Comment (RC1) · Laure Guerit (Referee) · 9 Dec 2019

**Review of Topographic controls on divide migration, stream capture, and diversification on riverine life, by Lyons et al, submitted to Earth Surface Dynamics.**

This paper is about the relationship between landscape evolution in response to base level fall or heterogeneous uplift and the evolution of species richness, based on a large number of numerical simulaitons. The authors use a free-access LEM to generate the landscape and develop a new component for the LEM to solve for species richness.
This work addresses very interesting questions on the links between perturbations, landscape and species richness. However, I found that the current form of the manuscript does not support this work as it should. The text is sometimes vague because of the use of generic words and absence of quantitative data, and some sentences are a bit complex and could be more straightforward. As a consequence, it is a bit difficult to follow the description and the arguments of the authors. I think the manuscript requires rewriting to clarify the context of this study, to ease the reading and to clearly support the purpose and the novelty of this work.

I hope my comments below can help,

Laure Guerit
Géosciences Environnement Toulouse, France

**Introduction**

First paragraph: I think the authors can present better what has been done before on drainage reorganization from field, lab and numerical studies. It seems that the real novelty of this work is the SpeciesEvolver they propose and the evolution of species within an evolving drainage network. This should be better presented and highlighted throughout the paper. In the current, this very interesting contribution is a bit lost among other things. Below is a small selection of papers that might be relevant for the general context and maybe elsewhere in the manuscript (sorry for the self-citation but it seems to be relevant for this paper. Note that I don't ask for reference to these papers, they are just some examples).

Second paragraph: add a reference at the end of line 25 to justify this statement or explain it a little bit here.

Third paragraph: the limits of the stream power model coupled to hillslope diffusion are discussed for quite some years (see for example Lague, 2014) and other models based on a different formalism have been proposed (see references below). As the choice of the model affects how the landscape responses to a perturbation (Armitage et al., 2018), this could be discussed in section 5.

**Description of modelling tools**

This section is too vague and it is difficult to get a correct idea of the numerical model used here. I suggest to be more specific, for example, name the fields, give the values, present the multiple components, etc. Also explain how the SolverEvolver is working: define what kind of species you are considering, how do you set the parameters, etc.

**Experiment design**

Here again, I suggest to be more specific and quantitative: what is the amplitude of the sea-level fall, of the uplift, how to you identify the variables and what are these variables (l.12). At the end of the first paragraph, you mention seven factors that are not listed below. Please name them and give range of values so that the following sections are easier to follow.

**Sensibility analysis**

Please clarify how you define the expected value of Y (l.4) and how the indices will be use in the following (end of the section).

**Model trial progression**

The values used in this study must be presented in the manuscript (at least in Supplementary Material)

**Initial conditions phase**

beginning of page 6: I don't understand how you generate the initial elevation grid. Please consider reformulate these sentences.

p.6 l.18 to 24 5mm/yr is also reported in New Zealand (eg, Jiao et al, 2017) while $10^{-5}$ corresponds to cratonic values. Maybe simply write that you consider uplift rates in the range of cratonic to orogenic values.

Additionnal references for erodibility and diffusion suggested below.

m and n: Kang and Parker (2018) suggest that the value of 0.5 should not be used as it leads to unrealistic behavior. Maybe the authors coudl run a few additional simulations to check whether they do observe the same behavior with m/n = 0.4 for example (this does not have to be part of the main manuscript).

p.7 l.12 describe or add a figure to illustrate.

**Perturb phase**

p.7 l.14 describe the steady state topography (for example the elevation and the number of catchments)
p.7 l.21 describe how the landscape responses to the perturbation. Is it only by knickpoint propagation ? What happens on the hillslopes ?
p.8 l.11 the way to define steady state could be recall here.

**Model response variables**

l. 13 what variables ?
p.9 l.1 the model descriptions must be within this manuscript.
p.9 l.4 specify what minimally implies
p.9 l.5 unclear, consider reformulate this sentence.
p.9 l.7 please give the size in meter
p.9 l.7 the sentence is odd with respect to the previous one saying that the streams are minimally affected. If so, why is the main divide migrating ?
p.9 l.9 a quantitative value or a figure to support this statement would be welcome.
p.9 l.12 «sufficiently» please quantify
p.9 l.16 please consider reformulate. This sentence suggests that they are two main divides (the main one and the main on the upthrow block), which is odd.

**Topographic relief and landform change**

The first paragraph is a bit complexe to follow, it could be written in a more straightforward way to ease the reading.
l.25 11 000 m seems high for a terrestrial landscape.
l. 29 the evolution of the topography is controlled by the stream power model (your equation 5). The main controlling factors are  U and K so I don't think the total order Sobol indices analysis is required here. This would simplify this section.
p. 10 l.3 please quantify «low relative»
p. 10 l.8 please quantify «high»
p. 10 l.10 could you add a figure to support this statement ?
p. 10 l.14 please quantify «low»
p. 10 l.17 please quantify «sufficiently high»
p. 10 l.23 please define what is a divide change

**Stream capture occurrence**

This section is more about the controls of the occurence than the occurence itself so the title could be adjusted to better reflect the content of this section.
p. 11 l.33 please quantify «moderately high»

**Species richness**

Here again, the section is more on the controls on the species richness than on the richness itself. The title should be adjusted to reflect the content of this section.
l.9 unclear, please consider reformulate
l. 16-22 this paragraph should come first in the section
p. 12 l.21 please specify  «less than» what ?

**Discussion**

p.13 l.8 a short description to the chi metric could be proposed here and a proper chi analysis could be performed to support the discussion.
p. 13 l.15 please quantify «greater increase»
p.13 l.17 define «a certain relief»
p.14 l.4 did you work with higher Pm values ? Does it influence this behavior ?
p.14 l.4 quantify «relatively high»
p.14 l.4 define what is an «elongated divide migration»
p.14 l.14 specify «more than » what
p14 l16 captures should be captured

**Conclusions**

As suggested for the introduction, it seems that the novelty of this work is the relationship between species richness and drainage reorganization rather than reorganization itself. This should be better highlighted here.
Can the authors comment on the value of 439% ? Is there a way to compare with natural landscape ?
This kind of quantification is missing in the rest of the paper to support the work of the authors.

**Table 2** considering the range of uncertainties, the statistics could be close to 0. Could the authors comment on that ?

**Figure 6c-d** missing labels

**Some references about drainage reorganization and chi**

- Bishop (2007) Long-term landscape evolution: linking tectonics and surface processes
- Bonnet (2009) Shrinking and splitting of drainage basins in orogenic landscapes from the migration of the main drainage divide
- Perron and Royden (2012) An integral approach to bedrock river profile analysis
- Guerit et al (2018) Landscape 'stress' and reorganization from chi-maps: Insights from experimental drainage networks in oblique collision setting

**Reference to landscape and species evolution** (with references inside that might be very relevant to this work)

- Salles et al (2019) Mapping landscape connectivity as a driver of species richness under tectonic and climatic forcings

**Reference to the stream power model** (and references therein)

- Lague (2014) The stream power river incision model: evidences, theory and beyond

**References to other models**

- Armitage et al. (2018) Numerical modelling of landscape and sediment flux response to precipitation rate change
- Carretier et al. (2016) Modelling sediment clasts transport during landscape evolution: Earth Surface Dynamics, v. 4, p. 237–251
- Shobe et al. (2017) The SPACE 1.0 model: A Landlab component for 2-D calculation of sediment transport, bedrock erosion, and landscape evolution: Geoscientific Model Development, v. 10, p. 4577–4604,
- Langston and Tucker (2018) Developing and exploring a theory for the lateral erosion of bedrock channels for use in landscape evolution models: Earth Surface Dynamics, v. 6, p. 1–27
- Yuan et al. (2019) A new efficient method to solve the stream power law model taking into account sediment deposition: Journal of Geophysical Research: Earth Surface
- Jiao, R., Herman, F., and Seward, D.: Late Cenozoic exhumation mo- del of New Zealand: Impacts from tectonics and climate, Earth- science reviews, 166, 286–298, 2017.
- Kwang and Parker (2018) Landscape evolution models using the stream power incision model show unrealistic behavior when m/n equals 0.5

**References to m/n, K, Kd**

- Whipple and Tucker (1999) Dynamics of the stream-power river incision model: Implications for heigh limits of mountain ranges, landscapes response timescales, and research needs
- Snyder et al. (2000) Landscape response to tectonic forcing: Digital elevation model analysis of stream profiles in the Mendocino junction region, northern California
- Wobus et al. (2006) Tectonics from topography: Procedures, promise, pitfalls
- Perron et al. (2009) Formation to evenly spaced ridges and valleys.

---

## Referee Comment (RC2) · Anonymous Referee #2 · 15 Mar 2020

General comments:

The authors present results from a new macroevolution model coupled with a landscape evolution model, examining how variation in geomorphological parameters drive drainage reorganization and, through drainage reorganization, speciation and extinction. Overall, the manuscript is clearly written and I was able to follow the authors' logic section to section. The problem of coevolution of drainage networks and the aquatic species that populate them is interesting and important and the authors' work on the SpeciesEvolver component is a strong contribution. If this manuscript is intended to introduce to the SpeciesEvolver model to geomorphologists and demonstrate an application alongside other Landlab tools, then it works pretty well. However, if the modeling results presented here are intended to say something substantive about the relationships between geomorphological parameters, drainage basin reorganization, and the evolution of species that inhabit them, I think there are some significant problems. First off, I think it needs to be more clearly stated whether the authors' goal is the former or the latter. If the goal is to say something meaningful about speciation and topography and not just "check out the cool experiments you can do with the tool we made", then there should either be some sort of field data incorporated (which would be really difficult) or some of the unrealistic conditions associated with these model runs need to be changed or at least convincingly addressed in the text.

More specific comments:

Page 3 Line 7: I don't think a landscape evolution model that neglects mass wasting will realistically represent divide migration where total relief is as high as it is in many of the simulations. I think it would be more meaningful to stick to relief ranges where diffusion could reasonably be assumed to be the dominant hillslope process if landslides aren't to be included.

Page 3 Line 21-Page 4 Line 6: The description of the SpeciesEvolver component needs more depth. The ESurf readership is going to be mainly geomorphologists. Speaking for myself, I hardly know anything about speciation and extinction and even less about the considerations involved in modeling these processes. It's an interesting tool and it deserves a lot more than two paragraphs included here. I don't understand very well how it works or why I can trust that it describes natural processes accurately.

Page 7 Line 5: Is it realistic for a species to occupy all parts of a stream network? The relief of some of the modeled landscapes described here definitely would give you different climate zones.

Page 7 Line 19: How is a perturbation of 0.1 m going to do anything to really modify the landscape, if we're interested in the divides? Along the same lines, why include

scenarios with a modeled fault displacement of 100 m when that's so much larger than anything observed in nature? If we're just trying to shake things up and see what happens, why keep other parameter values within empirically observed ranges?

Page 7 Line 22: Shouldn't knickpoints matter to the modeled species? Since knickpoint migration is what's transmitting the perturbation to the divides, I would think you'd need to account for the knickpoints' influence on aquatic life in order to accurately model what happens when the knickpoint makes it all the way upstream.

Page 8 Lines 6-7: Maybe I'm missing something, but why would D go extinct just because its river has been captured by C?

Page 8 Line 18: Does this mean that the divide percent change response only records whether a divide moved, and not how much it moved?

Page 9 Line 7: Will species in the north-draining rivers be more likely to go extinct due to loss of drainage/habitat area, or do they only go extinct when all drainage area is lost?

Page 9 Line 25: Why allow parameter combinations that lead to relief structures that are impossible to produce under Earth conditions? I just think it undermines the results a bit.

Page 10 Line 10-11: Does changed here just mean that they moved or that they were incorporated into a different drainage network?

Page 11 Line 5: Does 3% seem like a reasonable value compared to real landscapes? There seems to be evidence for stream capture all over the place. I think it would help me to understand better what's going on in the model landscapes if I had a better idea what the distribution of relief was. Maybe they're mostly very low.

Page 12 Line 1: Is this the formation of endorheic basins?

Page 13 Line 5: It doesn't seem like there are all that many landscapes that commonly

experience perturbations resulting from fault slip or base level fall where the perturbation magnitude approaches the relief magnitude. Again, I just wonder whether these model scenarios are realistic enough to provide meaningful insights in a lot of the iterations.

Minor nit-picks: Page 8 Line 16: Should say species diversification?

Page 12 Line 25: Reads better if the sentence doesn't begin with "Although"

Page 12 Line 29: This is the first time I've seen "lineage response" in this paper.

Page 13 Line 8: Should say "Cross-divide difference in relief"?

Page 13 Line 9: relief, thus

Page 13 Line 13: landscapes, although topographic relief

---

## Author Comment (AC1) · 12 Apr 2020

**Review of Topographic controls on divide migration, stream capture, and diversification on riverine life, by Lyons et al, submitted to Earth Surface Dynamics.**

This paper is about the relationship between landscape evolution in response to base level fall or heterogeneous uplift and the evolution of species richness, based on a large number of numerical simulaitons. The authors use a free-access LEM to generate the landscape and develop a new component for the LEM to solve for species richness.

This work addresses very interesting questions on the links between perturbations, landscape and species richness. However, I found that the current form of the manuscript does not support this work as it should. The text is sometimes vague because of the use of generic words and absence of quantitative data, and some sentences are a bit complex and could be more straightforward.

As a consequence, it is a bit difficult to follow the description and the arguments of the authors. I think the manuscript requires rewriting to clarify the context of this study, to ease the reading and to clearly support the purpose and the novelty of this work.

I hope my comments below can help,

Laure Guerit
Géosciences Environnement Toulouse, France

> Your comments certainly did help. Thank you for your time. Our responses to your inline comments below cover the topics in your introduction to your review.

**Introduction**

First paragraph: I think the authors can present better what has been done before on drainage reorganization from field, lab and numerical studies. It seems that the real novelty of this work is the SpeciesEvolver they propose and the evolution of species within an evolving drainage network. This should be better presented and highlighted throughout the paper. In the current, this very interesting contribution is a bit lost among other things. Below is a small selection of papers that might be relevant for the general context and maybe elsewhere in the manuscript (sorry for the self-citation but it seems to be relevant for this paper. Note that I don't ask for reference to these papers, they are just some examples).

> We agree the introduction should better highlight prior work in drainage reorganization along with species macroevolution and our key contribution of integrating the two through modeling. We added additional prior work context and references for both topics, while keeping paper length in mind, and emphasized our contribution more in the introduction, discussion, and conclusions.

Second paragraph: add a reference at the end of line 25 to justify this statement or explain it a little bit here.

> Improved our explanation of dispersal here.

Third paragraph: the limits of the stream power model coupled to hillslope diffusion are discussed for quite some years (see for example Lague, 2014) and other models based on a different formalism have been proposed (see references below). As the choice of the model affects how the landscape responses to a perturbation (Armitage et al., 2018), this could be discussed in section 5.

> In the revision we included a new section in the discussion regarding limitations of our model. We included how limitations of the stream power model and hillslope diffusion factor into our model.

**Description of modelling tools**

This section is too vague and it is difficult to get a correct idea of the numerical model used here. I

suggest to be more specific, for example, name the fields, give the values, present the multiple components, etc. Also explain how the SolverEvolver is working: define what kind of species you are considering, how do you set the parameters, etc.

> This section provides a general description of the tools used to build the model of the study; therefore, the goal of this section is not to explain the numerical model used in this study. The following section, 'Experiment design' presents our application of the tools to create a numerical model, including descriptions of components, fields, etc. We find it preferable to separate these because (1) readers are guided from general to more specific, (2) this organisation separates what we are using in the study (the tools) from our application (the model), and (3) including the level of detail of components, etc here would greatly increase the length of this section because the components, etc would have to again be put into the context of the study.
>
> In this revision, we explicitly indicate the purpose of the section and more succinctly describe the tools to make clear it is an overview of the tools.

**Experiment design**

Here again, I suggest to be more specific and quantitative: what is the amplitude of the sea-level fall, of the uplift, how to you identify the variables and what are these variables (l.12). At the end of the first paragraph, you mention seven factors that are not listed below. Please name them and give range of values so that the following sections are easier to follow.

> The absence of introduction to the seven factors was addressed by including in this paragraph/section (3.0) a reference to Table 1 and indicating the precise section where the factors are described. This table includes both the magnitudes of base level fall and fault throw as well as the other sensitivity analyses factors. The values of factors vary in experiment trials, so these cannot be quickly summarized without great redundancy with later sections. The inclusion of Table 1 provides the introduction to the factor names to help readers link this section introduction paragraph to later subsections.
>
> As this paragraph is an overview of all of section 3, description of the factors is held off until they come into play in the 3.2. Section 3.1 explains the sensitivity analyses providing how factors in general are used and why there is a range in factors.

**Sensibility analysis**

Please clarify how you define the expected value of Y (l.4) and how the indices will be use in the following (end of the section).

> Clarified that the expected value is more precisely the conditional expectation, in terms of probability theory. Our use of Sobol indices in identifying the most influential factors on a response was added to the end of this section.

**Model trial progression**

The values used in this study must be presented in the manuscript (at least in Supplementary Material)

> This comment is in response to the following text in the manuscript: "The factor values for each trial are available in Lyons et al. (2019)." This reference is the data repository that contains the factor values of the 51,200 trials for this study. Using a data repository such as Zenodo follows the recommendations of the journal. In the revision, we explicitly indicate that the reference is a data repository associated with the paper.

**Initial conditions phase**

beginning of page 6: I don't understand how you generate the initial elevation grid. Please consider reformulate these sentences.

> Two sentences were added for clarification beginning with, "The initial topography of each trial was generated in a two-step process...". The first paragraph containing the sentences in question describe how the initial random elevation values were set. The description of the initial elevation creation does not really begin until the second paragraph of this section. I find that the new sentences clarify the generation of the initial elevation values.

p.6 l.18 to 24 5mm/yr is also reported in New Zealand (eg, Jiao et al, 2017) while $10^{-5}$ corresponds to cratonic values. Maybe simply write that you consider uplift rates in the range of cratonic to orogenic values.

> This is a fantastic idea. (thank you!) We added this as a simplification and we retained some of our prior explanation for those less comfortable with the terms cratonic and orogenic, given that this paper may appeal to those with less of a geoscience background.

Additionnal references for erodibility and diffusion suggested below.

> We added an additional reference for erodibility. We sought references that were directly comparable, i.e., m/n is the same in the reference in our paper. For the diffusion coefficient, we cited a review paper with several references. Suggested references do not contain a comparable D, for example Perron et al. (2009) provides D/K ($m^{2m+1}$).

m and n: Kang and Parker (2018) suggest that the value of 0.5 should not be used as it leads to unrealistic behavior. Maybe the authors coudl run a few additional simulations to check whether they do observe the same behavior with m/n = 0.4 for example (this does not have to be part of the main manuscript).

> The paper that you describe, Kwang and Parker (2017) states, "when hillslope diffusion is neglected, the choice m/n=0.5 yields a curiously unrealistic result...". We did incorporate hillslope diffusion; therefore, this model limitation does not apply here.

p.7 l.12 describe or add a figure to illustrate.

> This comment refers to the following sentence: "Across the trials during this phase, factor values produced different initial stream networks and species locations." We removed this line that we now recognize is more of a result than a method.

**Perturb phase**

p.7 l.14 describe the steady state topography (for example the elevation and the number of catchments)

> The steady state topography is described in the results section because it is an outcome of the model factor combinations. Metrics of the tens of thousands of unique landscapes are summarized in Table 2. The model responses, including relief, of each trial is provided in the data repository reference, Lyons et al. (2019).

p.7 l.21 describe how the landscape responses to the perturbation. Is it only by knickpoint propagation ? What happens on the hillslopes ?

> We rewrote this sentence to be more direct in why this equation is presented. The landscape response to the perturbation is thoroughly described in the beginning of section 4. This description was improved in the revision.

p.8 l.11 the way to define steady state could be recall here.

Included steady state conditions here as well.

**Model response variables**

l. 13 what variables ?

In the revision we now recall the explanation of response variables directly under the header of this section.

p.9 l.1 the model descriptions must be within this manuscript.

This comment concerns a citation of Lyons et al. (2019), which is the dataset of this research in the Zenodo repository. It is now explicitly mentioned that the reference is a data repository associated with the paper.
* * *
The comments within these dashes were generally addressed by focusing more directly on exemplary model trials. Exemplary were also used in the prior version, although our prior explanations were unnecessarily confusing by attempting too much to generalize with all experiment trials when the exemplary trials often suffice.

p.9 l.4 specify what minimally implies

The streams in the lower grid are now described as remaining fixed, which is the case for the exemplary trials, rather than minimally shifting.

p.9 l.5 unclear, consider reformulate this sentence.

We reformulated this sentence for clarity.

p.9 l.7 please give the size in meter

A newly included measure of main divide migration enabled us to include 250 m instead of "a few nodes".

p.9 l.7 the sentence is odd with respect to the previous one saying that the streams are minimally affected. If so, why is the main divide migrating ?

The idea being no lateral stream erosion while streams erode headward. This paragraph was rewritten in the revision to clarify this.

p.9 l.9 a quantitative value or a figure to support this statement would be welcome.

Improved in the revision is clarification that comparison of the analytically-predicted and numerically-modelled knickponts is illustrated in supplementary animated videos. Animations include quantification of knickpoint propagation using Eq. 6.

p.9 l.12 «sufficiently» please quantify

"sufficiently" was quantitatively put into context of the minimum perturbation magnitude required for main divide migration now described earlier in this section.

p.9 l.16 please consider reformulate. This sentence suggests that they are two main divides (the main one and the main on the upthrow block), which is odd.

We interpret this comment as a misunderstanding of our intent to compare main divide migration in the two scenarios. We rewrote this sentence to clarify the nature of the main divide in the two scenarios.
* * *
**Topographic relief and landform change**

The first paragraph is a bit complexe to follow, it could be written in a more straightforward way to ease the reading.

> We have rewritten the paragraph for clarity.

l.25 11 000 m seems high for a terrestrial landscape.

> We included in the discussion that the maximum relief outputted in a trial is greater than observed, notably that mass wasting not included could contribute to the discrepancy. It is our opinion that the discrepancy is small especially given the simplicity of the model.

l. 29 the evolution of the topography is controlled by the stream power model (your equation 5). The main controlling factors are U and K so I don't think the total order Sobol indices analysis is required here. This would simplify this section.

> Perhaps those more familiar with the stream power model will understandably question the purpose of the analysis for these variables given that the control of U and K on relief is easy to understand given the simplicity of these variables in the equation. An intent of conducting the Sobol analysis on U and K in respect to relief is to allow readers to confirm their understanding of how the Sobol indices work, and that they do work, prior to using these indices in later sections on more complex relationships between factors and responses. Further, this analysis is to emphasis the primary influence of factors on relief, which is critical for later sections.

p. 10 l.3 please quantify «low relative»

> First, this sentence was rewritten with U and K (model inputs) instead of relief (model output) to indicate the control on divide and stream location change. Also, the paper puts forth that the relative values of U/K (or relief) vs perturbation is what matters. The values of the parameters relative to each other is more important than their absolute values. Text was added here to emphasize this.

p. 10 l.8 please quantify «high»

> Our response to the prior comment also applies to this comment.

p. 10 l.10 could you add a figure to support this statement ?

> A figure already exists. This sentence describes another detail about the figure referenced in the prior sentence. Text modified to help make this clear.

p. 10 l.14 please quantify «low»

> Maximum relief is now indicted for trials with < 30 % divide change.

p. 10 l.17 please quantify «sufficiently high»

> Rewritten to describe the relationship among the trials between perturbation magnitude/fault scarp and stream/divide location change.

p. 10 l.23 please define what is a divide change

> Stream and divide change, collectively referred to as landform change, is defined in detail in

the methods. In the revision, these terms are recalled early in this results section, "Topographic relief and landform change".

**Stream capture occurrence**

This section is more about the controls of the occurence than the occurence itself so the title could be adjusted to better reflect the content of this section.

Adjusted section title.

p. 11 l.33 please quantify «moderately high»

Rewritten to describe the relationship of relief with stream change and capture when Pm:relief near 1.

**Species richness**

Here again, the section is more on the controls on the species richness than on the richness itself. The title should be adjusted to reflect the content of this section.

Adjusted section title.

l.9 unclear, please consider reformulate

Rewritten for clarity.

l. 16-22 this paragraph should come first in the section

We agree and reorganized the beginning of this section.

p. 12 l.21 please specify «less than» what ?

"topographic" was inserted before relief and the sentence was rewritten for clarity.

**Discussion**

p.13 l.8 a short description to the chi metric could be proposed here and a proper chi analysis could be performed to support the discussion.

Chi analyses are especially useful in real landscapes (i.e. not modeled) where natural topographic complexity/roughness is great. In modeled landscapes, analyses of other metrics (e.g. relief) lead to similar interpretations. Whipple et al. (2017) found relief to be a reliable predictor of drainage divide migration, and relief is already central to this paper.

p. 13 l.15 please quantify «greater increase»

We could not find "greater increase" at this line or elsewhere in the document.

p.13 l.17 define «a certain relief»

"a certain relief" is rewritten as "a given relief" to indicate the relationship described in this heading varies by the relief of the landscape, where the landscapes and their relief vary in the experiment trials.

p.14 l.4 did you work with higher Pm values ? Does it influence this behavior ?

The Pm of trials ranged 0.1 to 100 m. The Pm influences the proportion of divides that migrated as Pm is less than initial relief. Rewritten for clarity.

p.15 l.4 quantify «relatively high»

*We rephased sentence to describe the relationship of kd and other parameters in this relationship as the relationship is of greater importance than absolute values in this instance.*

p.14 l.4 define what is an «elongated divide migration»

*We removed "elongated divide migration" in a rewrite of this sentence to better describe that divides migrated a greater distance when initial relief was less than the perturbation magnitude in a trial.*

p.14 l.14 specify «more than » what

*Added "more important than the fault throw scenario" to indicate the scenario where Ac had a greater influence on model output factors.*

p14 l16 captures should be captured

*We interpret this comment as a misunderstanding of the intended meaning that captures become increasingly more frequent as Ac decreases. We rewrote for clarity.*

**Conclusions**

As suggested for the introduction, it seems that the novelty of this work is the relationship between species richness and drainage reorganization rather than reorganization itself. This should be better highlighted here.

*We appreciate you ensuring that the novelty of the work is highlighted sufficiently. In the first submission, the relationship between species richness and drainage reorganization was highlighted in the second paragraph of the conclusion. The centrality of this topic in our contribution was made stronger.*

Can the authors comment on the value of 439% ? Is there a way to compare with natural landscape ?

This kind of quantification is missing in the rest of the paper to support the work of the authors.

*In the revision we describe the parameter and conditions of the run that put this great increase in richness. In our modeling we find singular values are likely less useful in comparison with natural landscapes as they are the outcome of the interaction of multiple factors.*

**Table 2** considering the range of uncertainties, the statistics could be close to 0. Could the authors comment on that ?

*We clarify that the plus/minus values in this table indicate the range of values outputted by trials of the model experiment.*

**Figure 6c-d** missing labels

*We included labels added in the revision. The labels were omitted in the first submission because the axes in c-d are the same as a-b. Consistency among the subplots will be clearer.*

**Some references about drainage reorganization and chi**

- Bishop (2007) Long-term landscape evolution: linking tectonics and surface processes
- Bonnet (2009) Shrinking and splitting of drainage basins in orogenic landscapes from the

migration of the main drainage divide
- Perron and Royden (2012) An integral approach to bedrock river profile analysis
- Guerit et al (2018) Landscape 'stress' and reorganization from chi-maps: Insights from experimental drainage networks in oblique collision setting

**Reference to landscape and species evolution** (with references inside that might be very relevant to this work)

- Salles et al (2019) Mapping landscape connectivity as a driver of species richness under tectonic and climatic forcings

**Reference to the stream power model** (and references therein)

- Lague (2014) The stream power river incision model: evidences, theory and beyond

**References to other models**

- Armitage et al. (2018) Numerical modelling of landscape and sediment flux response to precipitation rate change
- Carretier et al. (2016) Modelling sediment clasts transport during landscape evolution: Earth Surface Dynamics, v. 4, p. 237–251
- Shobe et al. (2017) The SPACE 1.0 model: A Landlab component for 2-D calculation of sediment transport, bedrock erosion, and landscape evolution: Geoscientific Model Development, v. 10, p. 4577–4604,
- Langston and Tucker (2018) Developing and exploring a theory for the lateral erosion of bedrock channels for use in landscape evolution models: Earth Surface Dynamics, v. 6, p. 1–27
- Yuan et al. (2019) A new efficient method to solve the stream power law model taking into account sediment deposition: Journal of Geophysical Research: Earth Surface
- Jiao, R., Herman, F., and Seward, D.: Late Cenozoic exhumation mo- del of New Zealand: Impacts from tectonics and climate, Earth- science reviews, 166, 286–298, 2017.
- Kwang and Parker (2018) Landscape evolution models using the stream power incision model show unrealistic behavior when m/n equals 0.5

**References to m/n, K, Kd**

- Whipple and Tucker (1999) Dynamics of the stream-power river incision model: Implications for heigh limits of mountain ranges, landscapes response timescales, and research needs
- Snyder et al. (2000) Landscape response to tectonic forcing: Digital elevation model analysis of stream profiles in the Mendocino junction region, northern California
- Wobus et al. (2006) Tectonics from topography: Procedures, promise, pitfalls
- Perron et al. (2009) Formation to evenly spaced ridges and valleys.

> We carefully considered which of the above references were appropriate, as well as other additional references, and added those that were appropriate. We thank you for compiling this list.

---

## Author Comment (AC2) · 12 Apr 2020

**General comments:**

The authors present results from a new macroevolution model coupled with a land- scape evolution model, examining how variation in geomorphological parameters drive drainage reorganization and, through drainage reorganization, speciation and extinc- tion. Overall, the manuscript is clearly written and I was able to follow the authors' logic section to section. The problem of coevolution of drainage networks and the aquatic species that populate them is interesting and important and the authors' work on the SpeciesEvolver component is a strong contribution. If this manuscript is intended to introduce to the SpeciesEvolver model to geomorphologists and demonstrate an application alongside other Landlab tools, then it works pretty well. However, if the modeling results presented here are intended to say something substantive about the relation- ships between geomorphological parameters, drainage basin reorganization, and the evolution of species that inhabit them, I think there are some significant problems. First off, I think it needs to be more clearly stated whether the authors' goal is the former or the latter. If the goal is to say something meaningful about speciation and topography and not just "check out the cool experiments you can do with the tool we made", then there should either be some sort of field data incorporated (which would be really diffi- cult) or some of the unrealistic conditions associated with these model runs need to be changed or at least convincingly addressed in the text.

We thank you for your review. We find it much improves the paper. Immediately below, we provide an overview of the primary changes in the revision in very large part motivated by your review.

In the introduction we improved the overview of past work, including multiple recent modeling studies on drainage reorganization. Parametrization of many of these studies, and maybe future ones, is done with limited exploration of the parameter space, i.e., 1 to a few values for each parameter. Formally exploring the space is a substantial effort. We put forth that our motivation in the drainage reorganization sensitivity analyses is to (1) provide guidance to future studies given that we do explore such a wide parameter space and (2) describe key relationships among inputs and outputs given the processes we modeled.

Also in the introduction, we expand on the outline that the intent is two-fold: (1) present an approach to study the evolution of life alongside landscape evolution which we do by demonstrating SpeciesEvolver, and (2) explore the parameter space of commonly used process models used to simulate drainage reorganization and identify key patterns on inputs and outputs.

In methods, we more fully describe that a large parameter space is explored to avoid us selecting arbitrary bounds on parameter limits and to not invalidate the sensitivity analysis sampling (and use the discussion more fully for limitations to our approach). Discussed below in specific comments as well, the trial with the greatest relief is not necessarily the outcome of an unreasonable combination of parameters, but instead is the outcome of processes not included (e.g., mass wasting). Processes not included is discussed separately. Further, limiting parameter values given predefined combinations with other parameters as a precondition to run that combination invalidates the unbiased nature of sampling the parameter space.

A new section of the discussion is devoted to the primary model limitations and future adaptions. Absence of mass wasting and lithologic heterogeneity, stream power incision, and broad species dispersal ability are among the limitations emphasized here. Conclusions stated throughout the paper on the relative relief and perturbation impact on drainage reorganization is emphasized as this relationship is controlled by the processes included in the model. We envision that the heavy lifting of the thousands of modeling runs, all data made available, will provide a starting point for additional complexities in future work.

**More specific comments:**

Page 3 Line 7: I don't think a landscape evolution model that neglects mass wasting will realistically represent divide migration where total relief is as high as it is in many of the simulations. I think it would be more meaningful to stick to relief ranges where diffusion could reasonably be assumed to be the dominant hillslope process if landslides aren't to be included.

> In the revision we describe more in the paper introduction and more fully in discussion-limitations how our approach is focused on the contribution of the fluvial component of drainage reorganization with minimally consideration (i.e., diffusion) of hillslope processes. Also in the discussion we indicate that our model likely underpredicts divide migration as relief increases. Additionally, more critical to the species modeling is stream capturing than divide migration.

Page 3 Line 21-Page 4 Line 6: The description of the SpeciesEvolver component needs more depth. The ESurf readership is going to be mainly geomorphologists. Speaking for myself, I hardly know anything about speciation and extinction and even less about the considerations involved in modeling these processes. It's an interesting tool and it deserves a lot more than two paragraphs included here. I don't understand very well how it works or why I can trust that it describes natural processes accurately.

> We agree and find it generally challenging to provide a deeper description without making the paper unreasonably long. For this reason we cite a short paper that strictly describes the SpeciesEvolver software where users can go for further details of the tool made for species on continents in general, whereas this paper under review provides the first use of the tool, which is for riverine species and drainage reorganization.

Page 7 Line 5: Is it realistic for a species to occupy all parts of a stream network? The relief of some of the modeled landscapes described here definitely would give you different climate zones.

> In the paper revision this good point is included in the discussion on model limitations/considerations. Given that most networks span the total relief (from boundary to divide) and including species distribution by climate zones, the outcome would be a greater number of speciations for the higher elevation species, less for the lower elevation species. Most importantly to the study at hand and given we are familiar with model functionality in different setups, we can predict our interpretations would be quite similar, in terms of which inputs had a greater impact on which outputs, even if we do make incorporate this modification. Richness would be increased even more, given the greater number of initial species – the absolute increase of richness is not central to this study.

Page 7 Line 19: How is a perturbation of 0.1 m going to do anything to really modify the landscape, if we're interested in the divides? Along the same lines, why include scenarios with a modeled fault displacement of 100 m when that's so much larger than anything observed in nature? If we're just trying to shake things up and see what happens, why keep other parameter values within empirically observed ranges?

> We are not only interested in divide migration, but total stream and divide percent change as these metrics are readily comparable across the two scenarios. Even the 0.1 m perturbation affected these responses, under some combinations with other parameters, although minimally. It is useful in sensitivity analyses to include extremes of parameter value ranges that may matter less to help make clear which parameters matter most relative to other parameters.

> The 100 m perturbation magnitude of base level and fault throw is over 1000 years given that this is our time step. Even over this time span this is large (about 5 1999 Jiji earthquakes given throw measured from that event). We emphasize in the revision that this magnitude is

motivated by knickzones of this magnitude (100 m) which is key to propagation of the erosional wave that drives drainage reorganization.

Page 7 Line 22: Shouldn't knickpoints matter to the modeled species? Since knickpoint migration is what's transmitting the perturbation to the divides, I would think you'd need to account for the knickpoints' influence on aquatic life in order to accurately model what happens when the knickpoint makes it all the way upstream.

I am aware of studies that consider knickpoint/waterfall influence on individuals within the same species. I am not aware of studies of knickpoints/waterfalls as they relate to creation of species. There is potential for unidirectional geneflow (fish flowing down waterfalls), having less of an impact on divergence of populations above and below falls. Including impacts on dispersal by knickpoints would be a reasonable avenue for future research.

Page 8 Lines 6-7: Maybe I'm missing something, but why would D go extinct just because its river has been captured by C?

This is addressed in the text in the caption of Figure 3, which is, "While N3 did extend into the watershed of N4, it did not overlap the stream nodes of the prior time step, therefore N3 did not capture N4 following the strict definition of capture in this study." D becomes extinct because it does not overlap/been captured by another stream in T1. N3 and N4 are off by 1 cell across the time steps. Side note: SpeciesEvolver can be adapted to be less precise, eg species can disperse if streams are close, and not precisely from a time step to the next, although in this study, stream overlap across time steps had to be precise.

Page 8 Line 18: Does this mean that the divide percent change response only records whether a divide moved, and not how much it moved?

This is correct regarding the divide percent change response. In the revision, main divide migration distance was also calculated to provide a quantified sense of migration.

Page 9 Line 7: Will species in the north-draining rivers be more likely to go extinct due to loss of drainage/habitat area, or do they only go extinct when all drainage area is lost?

Species in north-draining rivers are more likely to become extinct in the base level fall scenario as the main divide approaches the northern boundary. They will go extinct once the outlet at the northern boundary has a drainage area below the critical drainage area, if the species does not exist in a stream network elsewhere in the grid. If it does exist elsewhere, only the species population in that shrunken drainage disappears.

Page 9 Line 25: Why allow parameter combinations that lead to relief structures that are impossible to produce under Earth conditions? I just think it undermines the results a bit.

This would involve selecting a maximum uplift rate and/or minimum erodibility that produces some relief put forth as reasonable relief (Mt Everest?, or one selected for a landscape without landsliding?). In this contribution we preferred the inverse approach of using the near gamut of observed parameters because these are the model inputs, rather than preconditioning results. This is not to argue that selecting a narrower range is not reasonable to produce landscapes more appropriate to the processes, rather we took an approach to provide the full gamut, and now, describe its limitations in the discussion.

Page 10 Line 10-11: Does changed here just mean that they moved or that they were incorporated into a different drainage network?

Rephrased to clarify that we are referring to the stream percent change response. Also in the revision, we restate the meaning of this response earlier in this section.

Page 11 Line 5: Does 3% seem like a reasonable value compared to real landscapes? There seems to be evidence for stream capture all over the place. I think it would help me to understand better what's going on in the model landscapes if I had a better idea what the distribution of relief was. Maybe they're mostly very low.

> We agree the distribution of relief should be better described. We do so in the revision. The 3 % should not be translated as anything like 3 % of landscapes on Earth have captures. If one did make that argument, one would be implying that these parameters are exponentially distributed in space for the parameters that were exponentially sampled.

Page 12 Line 1: Is this the formation of endorheic basins?

> Yes. We added mention of "endorheic" in this sentence.

Page 13 Line 5: It doesn't seem like there are all that many landscapes that commonly experience perturbations resulting from fault slip or base level fall where the perturbation magnitude approaches the relief magnitude. Again, I just wonder whether these model scenarios are realistic enough to provide meaningful insights in a lot of the iterations.

> More important to relating the model to the real world is not the total landscape relief, but the relief upslope of the perturbation. Perturbations can be propagated up landscapes as local base level drops. A base level fall/fault slip may have little effect across the total landscape beyond a steepened "erosional wave" moving upslope over time, upslope drainages may become more susceptible to reorganization if the erosional wave decays slower than the upslope relief decreases.

**Minor nit-picks:**

Page 8 Line 16: Should say species diversification?

> Yes. We inserted 'species lineage' prior to 'diversification'.

Page 12 Line 25: Reads better if the sentence doesn't begin with "Although"

> Removed "Although".

Page 12 Line 29: This is the first time I've seen "lineage response" in this paper.

> Removed "lineage response" by rewriting sentence with only introduced terms.

Page 13 Line 8: Should say "Cross-divide difference in relief"?

> Yes. Corrected.

Page 13 Line 9: relief, thus

> Inserted missing comma after 'relief'.

Page 13 Line 13: landscapes, although topographic relief

> Combined two sentences at "landscapes" and "although".

---

## Referee Report (RR1)

This is my second review on the manuscript entitled «Topographic controls on divide migration, stream capture, and diversification in riverine life» by Nathan Lyons et al, submitted to Earth Surface Dynamics.

As a general comment, I think the manuscript has been greatly improved and most of my questions and comments have been addressed. However, I still have two main comments on the revised manuscript :

- My main concern is about the topography of the simulated landscapes. In the first review, I mentioned that 11 000 m of elevation is quite high for a terrestrial landscape. The authors argued that this is related to the simplicity of the equations used in LandLab and that the discrepancy is thus small. I'm not a user of Landlab myself but I've been working with several numerical models based on the same kind of equations. Such models are able to produce realistic landscapes in terms of topography but some combinaisons of parameters can lead to unrealistic terrestrial landscapes.

One conclusion of the article is that the relative magnitude of perturbation to relief limits the landscape susceptibility to reorganisation. The total relief thus seems to be of importance for the conclusions of this work.

Accordingly, I would appreciate that the authors explicitly state that they do not work on terrestrial landscapes, or that they limit their analysis to landscapes that are similar to Earth. Given the exceptionally large number of simulations performed for each scenario (which is really impressive), this should not affect the conclusions of this contribution.

- In line with my first review, I still think that the main contribution concerns the co-evolution of species together with drainage reorganisation and that these key and new results could be better emphasized, in particular in the title and abstract. In the current manuscript, I still miss a proper paragraph in the introduction dedicated to the mechanisms that drive drainage reorganisation or to the questions that are still open regarding this topic. Such a paragraph would introduce the novelty of the results presented in section 4.2 for example.

Below are some additional line-by-line comments:

p 2. l 30 Please correct this sentence as the work of Bonnet (2009) is not computational but experimental.

p 6 l 7 Because there is life in the model, could a different time step (shorter, longer) affect the results ?

l 7 l 7 this rate is not exceptionally high, please remove exceptionnaly

p 7 l 16-17 the sentence of m and n should appear first as it justifies the units of K.

p 7 l 20 and following : this paragraph is a bit complex to follow, please consider rephrasing.

p 8 l 14 please add the Berlin and Anderson worked on the Roan Plateau as the values given here might not be universal.

p 8 l 19 and following : this paragraph is a bit complex to follow, please consider rephrasing.

p 10 l 26 : see my general comment about the maximum elevation. Do you mean that all simulations reach 11 000 m in elevation ? Given that the modelled landscape is at maximum 20 km long, this is really high.

p 11 l 21 - and similar sentences - Please clarify 30% of what

Section 5.1 references and discussions with respect to previous works on this question would be welcome in this section to emphasize the novelty of this contribution (see also my second main comment).

Laure Guerit
Géosciences Rennes, France

---

## Author Response (AR2)

**Associate Editor Decision: Publish subject to minor revisions (review by editor)** (29 Jul 2020) by Sebastien Castelltort

Comments to the Author:

Dear authors,

Your manuscript has been reviewed a second time. Both reviewers agree that the paper has improved significantly and deserves publication in eSurf after some remaining adjustments. Please take in consideration both referee reports and comments that come with this decision.

I would add that the authors could, if they wish, enrich the coverage of previous work on drainage reorganization by adding references to authors such as Goren, Willett, Guerit and Shelef who have all recently performed work on this topic, experimentally and numerically.

> We included additional citations to drainage reorganization work, including authors Bishop, Goran, Willett, Guerit, Shelef, and Prince; we agree these enrich the paper.

Second, I remain unsure as to whether a maximum relief of 11'000m is, or not, problematic. It is not necessarily shocking since these are numerical landscapes, but it still main underlie some slightly extreme values in the choice of parameters. You could decide to explicitly say that some of the landscape results are extreme (perhaps not terrestrial), or to remove the simulations that produced such high topography (given the very large number of simulation performed, this would not modify the overall picture).

> To summarize the issue of maximum relief present in the model experiment, we have included a new paragraph about this issue in a section on model limitations (Sect 5.5) and why we find it does not impact our interpretation.
>
> We now emphasize in two places in the text that relief was greater than 8000 m in only 0.12 % of the simulations ('trials' in the text), and relief was less than 1000 m in 89 % of trials.
>
> Limiting certain combinations of parameters greatly complicates, and maybe invalidates, the commonly used and rigorous sensitivity analysis method we employed.
>
> The range and combinations of model factors are well supported. We find the issue to be that erosional processes and magnitudes are not fully representative of Earth's range. (Doing so would add more experiment factors, addition of factors eventually makes computation unfeasible.) The simulations with the greatest relief are produced with high uplift rate and low erodibility. As Dr. Guerit points out in the latest review, the uplift rate we used is not exceptionally high, and the erodibility value is well within published values. The anonymous referee in the first review aptly pointed out the absence of mass wasting in the model, which of course has a large impact on erosion. Further, precipitation rate was fixed throughout the research at 1 m/yr, where greatly rates are found in some regions, e.g. the Olympic Mountains.

I'm checking the minor revision box, and I will look myself at your resubmission, i.e. the paper will not go for a third round of reviews.

All the best and thanks for your work.

SC

> Many thanks to you and reviewers for a diligent review.

This is my second review on the manuscript entitled «Topographic controls on divide migration, stream capture, and diversification in riverine life» by Nathan Lyons et al, submitted to Earth Surface Dynamics.

As a general comment, I think the manuscript has been greatly improved and most of my questions and comments have been addressed. However, I still have two main comments on the revised manuscript :

- My main concern is about the topography of the simulated landscapes. In the first review, I mentioned that 11 000 m of elevation is quite high for a terrestrial landscape. The authors argued that this is related to the simplicity of the equations used in LandLab and that the discrepancy is thus small. I'm not a user of Landlab myself but I've been working with several numerical models based on the same kind of equations. Such models are able to produce realistic landscapes in terms of topography but some combinaisons of parameters can lead to unrealistic terrestrial landscapes.

One conclusion of the article is that the relative magnitude of perturbation to relief limits the landscape susceptibility to reorganisation. The total relief thus seems to be of importance for the conclusions of this work.

Accordingly, I would appreciate that the authors explicitly state that they do not work on terrestrial landscapes, or that they limit their analysis to landscapes that are similar to Earth. Given the exceptionally large number of simulations performed for each scenario (which is really impressive), this should not affect the conclusions of this contribution.

> Few model trials had relief greater than Earth's modern maximum elevation. As an outcome of the factor sampling scheme described in Sect 3.1, relief was greater than 8000 m in only 0.12 % of trials, and relief was less than 1000 m in 89 % of trials, as represented in Fig. 4. In Sect. 4.1 near the beginning of the results section, we refer readers to Sect. 5.5 where we included a new paragraph on the trials with unrealistic relief and the affect they have on this research.

> We have also experienced that some factor combinations are problematic. Our prior point on model simplicity was that the model does not include mass wasting and diffusion is linear, both if included could produce more erosion. Also important and not clear in prior versions, precipitation was constant in all trials, and throughout space and time in all trials, at a rate of 1 m/yr. (We also now include the precipitation rate in Sect. 3.2.1.) The Olympic Mtns, for example have annual rainfall rate of ~5 m/yr, which of course greatly impacts erosion. In future research, model trials could vary rainfall rate. Each model factor varied in the sensitivity analyses greatly increasing the computation involved, up to an untenable amount.

- In line with my first review, I still think that the main contribution concerns the co-evolution of species together with drainage reorganisation and that these key and new results could be better emphasized, in particular in the title and abstract. In the current manuscript, I still miss a proper paragraph in the introduction dedicated to the mechanisms that drive drainage reorganisation or to the questions that are still open regarding this topic. Such a paragraph would introduce the novelty of the results presented in section 4.2 for example.

> The first paragraph of the introduction was revised to allow a more in-depth introduction to drainage reorganization. This was accomplished by moving text on biology to the next paragraph and adding in more prior research on the mechanisms that drive drainage reorganisation pertinent to the paper. Additional citations to this prior research was added in later sections as well.

> We sincerely appreciate your concern on emphasizing the main contribution of this research. We further emphasized the evolution of species following drainage reorganization in revisions to the abstract, a new paragraph in the introduction (the 2$^{nd}$ paragraph), a new brief section in discussion (5.4), and conclusions.

Below are some additional line-by-line comments:

p 2. l 30 Please correct this sentence as the work of Bonnet (2009) is not computational but experimental.

Thank you for pointing this issue out, specifically that this paragraph should present only computational research. We removed the citation to Bonnet (2009) at this location in the text.

p 6 l 7 Because there is life in the model, could a different time step (shorter, longer) affect the results ?

The short answer is it depends on how a different time step is implemented, and we effectively did present and discuss these results through the 'time to allopatric speciation' model factor.

The species in this first use of this modeling software directly and exclusively evolves in response to topographic change. Therefore, the impacts on life by different time step duration occurs only because of any effects of step duration on surface processes. On a different but related model approach, using a different time step than the surface processes for biological evolution was effectively investigated in this study through the "time to allopatric speciation" model factor that varied in experiment trials. Varying this factor is better than simply varying a time step duration for biological processes in totality because this parameter allows species to disperse in sync with topographic change while allowing for speciation operating at a different time scale than appropriate for surface processes. We described the time to allopatric speciation parameter in 3.2.1 and 3.2.2, as well as its impact on results in 4.3, and given the results, in the discussion we put forth that this process parameter can be broadly use to represent "speed at which species evolve" in 5.3. Detail is provided in the software documentation.

l 7 l 7 this rate is not exceptionally high, please remove exceptionnaly

We removed the word, exceptionally.

p 7 l 16-17 the sentence of m and n should appear first as it justifies the units of K.

We moved this sentence as suggested.

p 7 l 20 and following : this paragraph is a bit complex to follow, please consider rephrasing.

The content of this paragraph was previously stream identification and populating the model grid with species. Sentences switched between the two topics. We reorganised and rephrased such that first stream identification is described followed by a separate paragraph describing the species. Also, the sentences about the allopatric speciation parameter were moved to 3.2.2 where this parameter is also discussed. Moving the initial description previously in 3.2.1 to 3.2.2 is clearer because this is where this parameter is more in context.

p 8 l 14 please add the Berlin and Anderson worked on the Roan Plateau as the values given here might not be universal.

We added "on the Roan Plateau" to this sentence.

p 8 l 19 and following : this paragraph is a bit complex to follow, please consider rephrasing.

We rephrased several sentences in this paragraph and split the paragraph into two paragraphs.

p 10 l 26 : see my general comment about the maximum elevation. Do you mean that all simulations reach 11 000 m in elevation ? Given that the modelled landscape is at maximum 20 km long, this is really high.

Relief was different in each simulation ('trial' in the paper). We rephased this sentence. Above in this reply we discuss why we find relief not to meaningful affect our conclusions.

p 11 l 21 - and similar sentences - Please clarify 30% of what

Sentence rewritten as "The percentage of divides that changed location during the perturb phase reached only about 30 %...".

Section 5.1 references and discussions with respect to previous works on this question would be welcome in this section to emphasize the novelty of this contribution (see also my second main comment).

> We included additional references in Sect. 5.1 and especially more in later discussions sections where more citations were particularly helpful.

Laure Guerit
Géosciences Rennes, France

**Anonymous Referee #2**

Suggestions for revision or reasons for rejection (will be published if the paper is accepted for final publication)

The authors have made changes that have substantially strengthened the manuscript in my view, I just noticed a few awkward sentences at points that might benefit from being reworded, located at:

p. 13 line 25 - Stream network or all the networks of a species

Revised sentence: "Extinction in this simple implementation of SpeciesEvolver occurred only when all of the stream networks of a species disappeared."

p. 15 line 28 - kd was reached near the experiment maximum of this factor

"reached" was an unnecessary word here. Revision: "A greater proportion of divides changed location in the trials where steady state relief exceeded $P_m$, $k_d$ was near the experiment maximum factor value of $10^{-1}$ m$^2$ yr$^{-1}$…"

p. 16 lines 3-5 - The importance of stream locations would decrease in a landscape following multiple faults where more streams would more likely be near a fault.

Revised sentence: "The influence of the seed value and stream locations on drainage reorganisation would decrease relative to the other model factors in a landscape with multiple faults because more streams would more likely be near a fault."

Other than those few nit-picks, this is interesting work and the write-up is good. I'm looking forward to seeing how SpeciesEvolver is used moving forward in biogeomorphology!

Many thanks for the helpful comments and well wishes on SpeciesEvolver software.

[revised manuscript text omitted]

---

## Author Response (AR3)

Dear authors,

I'm pleased to confirm acceptance of your paper for publication in e-surf, after reviewing your revised manuscript in the context of the reviews and Associate Editor recommendation. I think your work will make a great addition to the journal and to the developing understanding of topographic and geomorphic controls on biological evolution. It will be exciting to see the directions that this model is taken in the future.

> We are pleased to hear about the contribution of this work to e-surf, and we appreciate the encouragement.

I've noted a few minor suggested edits for clarity and grammatical refinement, on the attached PDF. I leave these to your discretion, but you might consider attending to those change you'd like to make now, before uploading final versions for typesetting.

> We have addressed all suggested changes. Variations between the suggested changes and the final submitted document are minor and are detailed below. (Page and line numbers are those of the final document.)
>
> p1, l25: The following clause was removed because it is also in the next sentence: "especially for the species that evolved more rapidly".
>
> p2, l9: We did not include a comma as suggested as we believe our intended meaning was not clear. We rephrase portions of this sentence to clarify.
>
> p2, l20: We included "idea that" and also added "and frequent" as this a key finding in the cited paper that was erroneously omitted from the prior submission.
>
> p11, l22-24: prior to the correction about relief greater than observed, the order of sentence clauses was swapped to follow the context order of prior sentences (relief less than, followed by relief greater than).
>
> p17, l3-5: in addition to changing "as well as" to "and", we better separated each item of the list in this sentence for clarity.
>
> p17, 8-9, in addition to changing "that this" to "to be the", we removed repeating the word "factor", which necessitated emphasizing that we meant of the factors in this research.
>
> Table 1: The 'initial topography seed' does not have units.
>
> Corrections to references formatting.
>
> Figure 11, right subplot erroneously read "1 kyr: base level fall" instead of "1 kyr: fault throw". The latter is correct for this scenario/subplot, so this figure was corrected.

Also, please note that e-surf provides the scope to add links to data and videos as "Assets" associated with your paper. You can see this in some of the published articles. You might consider adding the

links to your videos that way. The Copernicus editorial team can help you with that.

> Our email including esurf editors and Copernicus on 3 Sep, 2020 determined the paper currently follows journal conventions, with no changes needed.

Thank you again for choosing e-surf for this work! And I am sorry that the review process on this submission was somewhat delayed.

Josh

> We appreciate your attentive help during the review process.

[revised manuscript text omitted]

**Commented [LNJ1]:** This figure was corrected, as pointed out by the editor, to include missing units in the x-axis label.

**Figure 4. Histogram of topographic relief.** The plotted data is the topographic relief at the trial end of the initial conditions phase of each trial. Note the y-axis is logarithmic.

[Figure]

**Figure 5. Sobol indices of topographic relief**. (a) The first and total order Sobol indices of relief at the initial steady state. Model input factors are on the x-axis where seed is the initial elevation seed. (b) Second order Sobol indices of relief. Factors are on the x- and y-axes. (c) Relief versus $U$ and $K$. Each point represents one of the unique steady state landscapes created in the initial conditions phase.

[Figure]

**Figure 6. First and total order Sobol indices of drainage reorganisation responses.** The factors are along the x-axis for each of the responses (a–h) where seed refers to the initial elevation seed.

[Figure]

**Figure 7. Landform change responses versus initial relief.** Responses of all trials for divide percent change (a–d) and stream percent change (e–f). The labelled points are the IDs of the exemplary trials depicted in Fig. 8 and described in the text.

[Figure]

**Figure 8. Landform change of exemplary trials.** This figure illustrates landform change model responses of the trials discussed in Sect. 4 and labelled in Fig. 7. The colour of grid cells symbolises landform type at the initial and final steady state in the model. Blue areas were not the landform type (divide or stream) in a given subplot at the times of either steady state. Red areas were the subplot landform type in both steady state times. Cyan and yellow areas were the subplot landform type in the initial and final steady state, respectively. The parameter values of these trials and all other trials are provided in Lyons et al. (2019).

[Figure]

Figure 9. Second order Sobol indices. Second order indices of paired model factors for the perturb phase responses. A relatively large value in a subplot indicates that the interaction of the factor pair affects the response more than other factor pairs with lower index values.

[Figure]

Figure 10. Capture count versus the ratio of $P_m$ and relief. $P_m$ and relief are equal at the vertical dashed line.

[Figure]

**Figure 11. Species richness percent change.** Species richness change versus relief (a–b) and capture count (c–d).

[Figure]

**Commented [LNJ2]:** right subplot erroneously read "1 kyr: base level fall" instead of "1 kyr: fault throw". The latter is correct for this scenario/subplot, so this figure was corrected

5  **Figure 12. Phylogeny of exemplary trials.** Topography was perturbed by base level fall of fault throw at 1 kyr elapsed since the first steady state was reached. Most of the trials animated in the video supplement are shown. Trial 5043 is not included. Species did not change in this trial because no stream networks disappeared or were captured. The lineages of clades in a trial are labelled alphabetically. Speciation events occurred where lineages split and extinctions occurred where lineages terminated before the end of the trial.